# Commensal bacteria maintain a Qa-1[b]-restricted unconventional CD8[+] T population in gut epithelium

**Jian Guan[1,2]\*, J David Peske[1,2], Michael Manoharan Valerio[3], Chansu Park[1,2], Ellen A Robey[3†], Scheherazade Sadegh-Nasseri[1]\*†**

[1]Department of Pathology, Johns Hopkins University School of Medicine, Baltimore, United States; [2]Institute of Cell Engineering, Johns Hopkins University School of Medicine, Baltimore, United States; [3]Division of Immunology and Molecular Medicine, Department of Molecular and Cell Biology, University of California, Berkeley, Berkeley, United States

**\*For correspondence:**
jian.guan.cc@outlook.com (JG);
ssadegh@jhmi.edu (SS-N)

†These authors contributed equally to this work

**Competing interest:** The authors declare that no competing interests exist.

**Abstract** Intestinal intraepithelial lymphocytes (IELs) are characterized by an unusual phenotype and developmental pathway, yet their specific ligands and functions remain largely unknown. Here by analysis of QFL T cells, a population of CD8[+] T cells critical for monitoring the MHC I antigen processing pathway, we established that unconventional Qa-1[b]-restricted CD8[+] T cells are abundant in intestinal epithelium. We found that QFL T cells showed a Qa-1[b]-dependent unconventional phenotype in the spleen and small intestine of naïve wild-type mice. The splenic QFL T cells showed innate-like functionality exemplified by rapid response to cytokines or antigens, while the gut population was refractory to stimuli. Microbiota was required for the maintenance, but not the initial gut homing of QFL T cells. Moreover, monocolonization with *Pediococcus pentosaceus,* which expresses a peptide that cross-activated QFL T cells, was sufficient to maintain QFL T cells in the intestine. Thus, microbiota is critical for shaping the Qa-1[b]-restricted IEL landscape.

## eLife assessment

This is an **important** study that investigates the role of commensal microbes and molecules in the antigen presentation pathway affecting the development and phenotype of an unusual population of T lymphocytes. The authors provide **compelling** evidence to identify a population of unconventional T cells that exist in the small intestinal epithelium, which appear to depend on commensal microbes, and show that a single commensal microbe (that encodes an antigen capable of weakly stimulating these cells) is sufficient to maintain this T cell population.

## Introduction

The display of peptides by MHC class I (MHC I) molecules on the cell surface is critical for CD8[+] T cell immune surveillance (*Shastri et al., 2002*). Generation of a peptide repertoire which accurately reflects intracellular events, such as viral infection or mutations, relies on a functional antigen processing and presentation pathway. After cytosolic cleavage of protein precursors and transport of peptides through the peptide transporter associated with antigen processing (TAP) into the endoplasmic reticulum (ER), the peptide intermediates are further customized by ER aminopeptidase associated with antigen processing (ERAP1) until 'ideal' peptides that fit the groove of MHC I molecules are eventually shuttled and displayed on the cell surface (*Serwold et al., 2002*; *Shastri et al., 2005*). Disruption of each step of the pathway can lead to immunological dysfunction (*Grandea*

*et al., 2000*; *Van Kaer et al., 1994*; *Van Kaer et al., 1992*). The critical role of ERAP1 in the antigen processing pathway has been established through extensive studies of ERAP1 deficient (ERAP1-KO) cells and mice (*Blanchard and Shastri, 2008*; *Guan et al., 2021*; *Hammer et al., 2007b*). Loss of ERAP1 severely disrupts the peptide repertoire presented by both the classical MHC Ia and the nonclassical MHC Ib molecules (*Hammer et al., 2006*; *Hammer et al., 2007a*). Qa-1[b], a nonclassical MHC Ib molecule, has been shown to present a significantly increased number of peptides on the cell surface of ERAP1-KO cells as compared with wild-type (WT) cells (*Nagarajan et al., 2016*). One peptide presented by Qa-1[b], FYAEATPML (FL9) (with the Qa-1[b]-FL9 complex termed QFL), was identified as an immunodominant ligand uniquely presented on ERAP1-deficient cells (*Nagarajan et al., 2012*). The CD8[+] T cells that specifically recognize this ligand are thus named QFL-specific T (QFL T) cells.

Early analysis of QFL T cells revealed the unusual nature of these CD8[+] T cells. Unlike conventional antigen-specific CD8[+] T cells which are typically detected at a frequency of 1 in $10^5$~$10^6$, QFL T cells are present at a frequency 10-fold higher in the spleen of naïve mice with the bulk splenic QFL T population displaying a CD44[hi]CD122[+] antigen-experienced phenotype (*Nagarajan et al., 2012*). T cell receptor (TCR) analysis of QFL T cells revealed that a large proportion of the QFL T population expresses an invariant TCRα-chain Vα3.2Jα21 (*Guan et al., 2017*). These traits of QFL T cells indicate their potential similarity to other unconventional T cells, such as invariant NKT (iNKT) and mucosal-associated invariant T(MAIT) cells, which are typically characterized by recognition of ligands presented by non-classical MHC Ib, expression of invariant TCRs, residence in nonlymphoid tissues and innate-like functions (*Godfrey et al., 2015*; *Salio et al., 2014*). The tissue distribution and functions of QFL T cells remain to be fully elucidated.

The gut mucosa is an immunologically complex niche with abundant lymphoid populations (*Faria et al., 2017*). The small intestinal intraepithelial lymphocyte (siIEL) compartment is populated by unconventional T cells of both TCRγδ[+] and TCRαβ[+] lineage which express CD8αα but lack CD4, CD8αβ, CD5, and CD90 expression. The development of the CD8αα[+]CD4[-]CD8αβ[-]TCRαβ[+] population (CD8αα[+] IEL), categorized as natural IELs (natIELs) because they acquire their activated phenotype in the thymus, has been extensively studied (*Cheroutre et al., 2011*). Yet little is known about their antigen specificity and TCR repertoire. Emerging evidence shows that nonclassical MHC Ib molecules are important for the development and effector function of CD8αα[+] IELs (*Das and Janeway, 2003*). For instance, while the loss of classical MHC Ia molecules showed little impact on the CD8αα[+] IEL population, these cells were decreased in mice deficient for Qa-2 (*Das et al., 2000*; *Das and Janeway, 1999*; *Park et al., 1999*). Furthermore, a population of CD8αα[+] IEL precursors which preferentially expresses Vα3.2 was shown to be decreased in number in the absence of CD1d (*Ruscher et al., 2017*). However, whether Qa-1[b] is involved in shaping the CD8αα[+] IEL population remains to be studied. In addition to MHC molecules, gut microbiota plays a critical role in the establishment and shaping of the gut immune system. Studies of germ-free (GF) mice have shown extensive defects in gut-associated lymphoid tissues together with morphological changes in the intestine associated with the absence of gut microbiota (*Round and Mazmanian, 2009*). Notably, despite the critical role of gut microbiota in the establishment of the gut immune system, natIELs that do not rely on cognate antigens in the periphery are believed to be home to the gut independent of microbes (*Mota-Santos et al., 1990*).

Here, we found abundant unconventional QFL T cells in both the spleen and siIEL compartment of naïve WT mice. The splenic Vα3.2[+]QFL T cells expressed high levels of CD44 and showed typical innate-like functions including hyperresponsiveness to cytokines and rapid IFN-γ production in response to antigen. In contrast, the populations of the same antigen specificity in the gut phenotypically resembled natIEL and were likewise functionally quiescent. Analysis of mice deficient of molecules associated with Qa-1[b]-FL9 antigen presentation revealed that TAP was required for the presence of the splenic or siIEL QFL T cells, whereas Qa-1[b] was essential for imprinting their unconventional phenotype. Analysis of GF mice showed that gut microbiota was needed for the long-term maintenance of QFL T cells. Furthermore, we found that maintenance of the gut Vα3.2[+]QFL T was associated with colonization by the commensal bacterium *Pediococcus pentosaceus* (*P. pentosaceus*) which expresses an FL9 homolog that could cross-activate QFL T cells. Overall, these results establish Vα3.2[+]QFL T cells as a unique population of unconventional CD8[+] T cells that rely on nonclassical MHC Ib to acquire their phenotype, and gut microbiota to be properly maintained in the intestine.

## Results

### Vα3.2$^+$QFL T cells are abundant in the spleen and gut of naïve wild-type mice

To identify QFL T cells in tissues of naïve WT mice, we generated Qa-1$^b$-FL9 dextramers (QFL-Dex) based on the 'dextran-doping' technique (*Bethune et al., 2017*). Prior work of our lab showed that unlike the Qa-1$^b$-Qdm tetramer which stains NK cells, the Qa-1$^b$-FL9 tetramer (QFL-Tet) does not bind to the NKG2A receptor but specifically stains CD8$^+$ T cells (*Nagarajan et al., 2012*). Because the same QFL monomers were used for generating the QFL-multimers in this study, we reasoned that it is unlikely that the QFL-Dex will exhibit shifted specificity. Moreover, in comparison with QFL-Tet, QFL-Dex showed improved sensitivity and specificity for QFL T cell detection (*Figure 1—figure supplement 1*), and further allowed enrichment of QFL T cells using magnetic beads. The average number of QFL T cells – defined as CD45$^+$CD19$^-$TCRβ$^+$CD4$^-$QFL-Dex-PE$^+$APC$^+$ (*Figure 1a*) – detected in the siIEL compartment of naïve WT mice was comparable to the population in the spleen. Additionally, QFL T cells were present at a frequency of ~1 in 1000 of the siIEL CD8$^+$ T cells, which was 10 times more frequent than in the spleen (*Figure 1b and c*). Both the splenic and siIEL QFL T cells were essentially all CD8$^+$ T cells, as these cells were barely detectable within the CD4$^+$ T population (*Figure 1—figure supplement 2*). Consistent with prior studies, ~80% of the splenic QFL T cells expressed the 'invariant' TCR Vα segment Vα3.2 which was significantly higher than the proportion of Vα3.2-expressing cells within the total CD8$^+$ T population in the spleen (<5%) (*Figure 1d and e*; *Guan et al., 2017*). Notably, QFL T cells in the siIEL compartment also contained a relatively larger proportion of Vα3.2$^+$ cells (20.1%) compared to the total CD8$^+$ TCRαβ siIELs (<10%) (*Figure 1f and g*). The average percentage of Vα3.2$^+$ cells within the QFL T population (20.1%) was lower in the siIEL compartment than in the spleen (63.3%). Vα3.2$^+$ QFL T cells were nevertheless abundantly present in both the spleen and gut of naïve WT mice.

### The unconventional phenotype of Vα3.2$^+$QFL T cells

We have previously observed a high percentage of CD44$^{hi}$ cells within the bulk splenic QFL T population of naïve WT mice (*Guan et al., 2017*; *Nagarajan et al., 2012*). Here, we further investigated the association between preferred Vα3.2 usage by QFL T cells and their phenotype. We observed that Vα3.2$^+$QFL T cells contained a significantly larger proportion of CD44$^{hi}$ cells (~90%) than the Vα3.2$^-$QFL T cells, suggesting that the antigen-experienced phenotype of the QFL T population was mainly contributed by the Vα3.2-expressing subpopulation (*Figure 2a and b*). In contrast, only around 20% of the total Vα3.2$^+$CD8$^+$ T population expressed high CD44. It is worth noting that the percentage of CD44$^{hi}$ cells within the total Vα3.2$^+$CD8$^+$ T population was slightly higher compared to the Vα3.2$^-$ population (*Figure 2—figure supplement 1a*), which was in line with a previous report (*Prasad et al., 2021*). We thus conclude that although both Vα3.2$^+$ and Vα3.2$^-$ cells were detected within the splenic QFL T population, a memory phenotype was specifically enriched in Vα3.2$^+$QFL T cells.

Given the high frequency of QFL T cells in the gut, we further investigated their phenotype. We assessed CD8αα expression, a hallmark of the unconventional phenotype of natIELs, on Vα3.2$^+$ and Vα3.2$^-$ QFL T cells in the siIEL compartment of naïve WT mice. A relatively larger fraction of Vα3.2$^+$QFL T cells lacked CD4 and CD8αβ expression but expressed CD8αα compared to the Vα3.2$^-$ subpopulation suggesting that the majority of Vα3.2$^+$ QFL T cells were CD8αα$^+$ natIELs (*Figure 2c and d*). No such difference was observed between the total Vα3.2$^+$ and Vα3.2$^-$ CD8$^+$ IEL populations (*Figure 2—figure supplement 1b*). These observations indicate that the Vα3.2$^+$ QFL T cells in both the spleen and siIEL compartment are unconventional CD8$^+$ T cells which likely are derived from the same precursors in the thymus.

### The impact of Qa-1$^b$, ERAP1, and TAP on Vα3.2$^+$QFL T cells

We further investigated what drives the unconventional phenotype of Vα3.2$^+$ QFL T cells by analyzing mice deficient in molecules that impact Qa-1$^b$-FL9 antigen presentation including Qa-1$^b$, ERAP1, or TAP (Qa-1$^b$-KO, ERAP1-KO, TAP-KO). While we consistently observed loss of both the splenic and siIEL Vα3.2$^+$QFL T cells in TAP-KO mice, the populations were detectable in both tissues of Qa-1$^b$-KO or ERAP1-KO mice with a slightly reduced population size in the spleen of Qa-1$^b$-KO mice (*Figure 3a*). Vα3.2$^+$ QFL T populations showed reduced percentages of CD44$^{hi}$ cells in the spleen of Qa-1$^b$ and

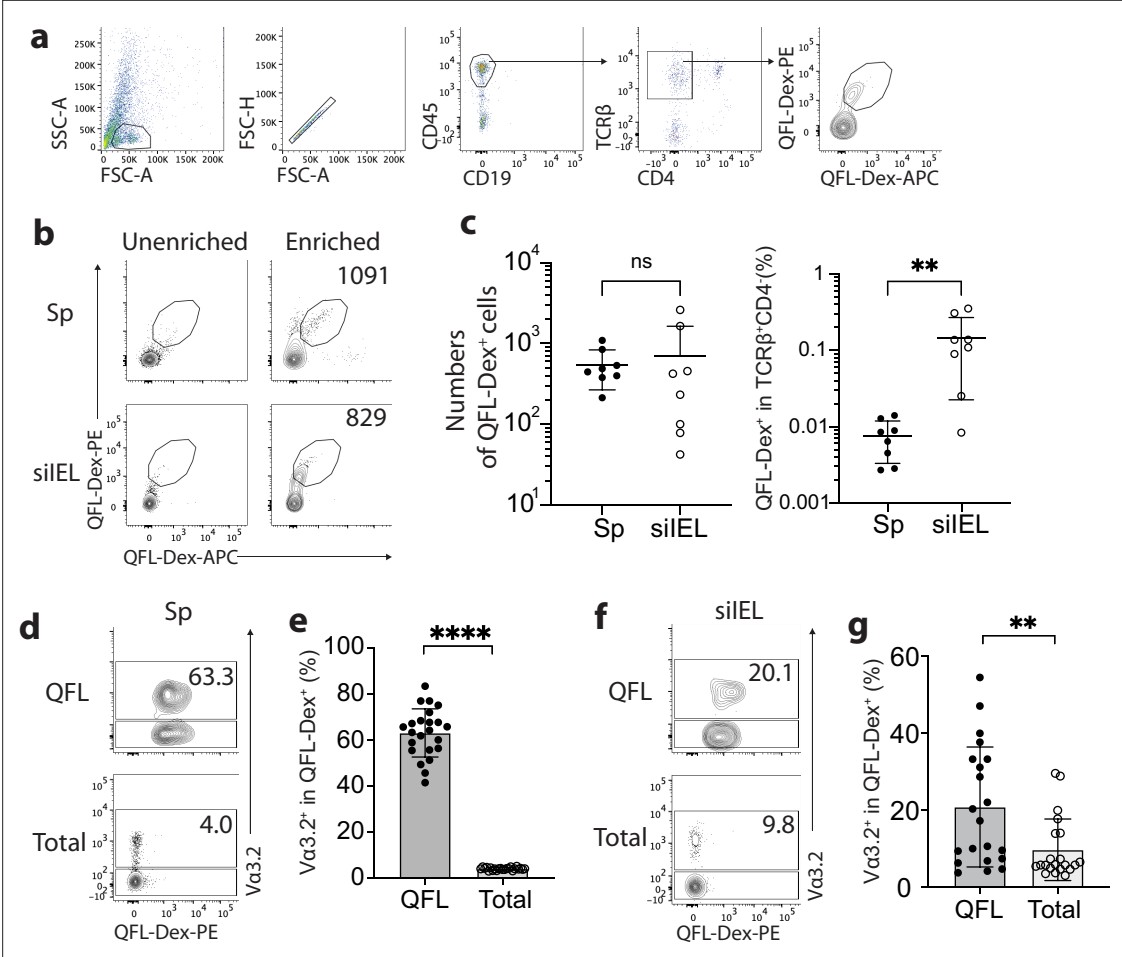

**Figure 1.** Abundant Vα3.2⁺ QFL T cells in both the spleen and small intestinal intraepithelial lymphocyte (siIEL) compartment of naïve wild-type (WT) mice. (**a**) Definition of QFL T cells by flow cytometry. Cells from the spleen or siIEL compartment were stained with Qa-1ᵇ-FL9 (QFL) dextramers labeled with phycoerythrin (QFL-Dex-PE) or allophycocyanin (QFL-Dex-APC) and analyzed for before (Unenriched) or after (Enriched) magnetic enrichment of dextramer-positive cells. QFL T cells are defined as the CD45⁺CD19⁻TCRβ⁺CD4⁻QFL-Dex-PE⁺APC⁺ population. Plots representing siIELs from naïve WT mice after enrichment for QFL-Dex⁺ cells. (**b**) Flow cytometry of cells from the spleen (Sp) and siIEL compartment of naïve WT mice 'Unenriched' or 'Enriched' for dextramer-positive cells. Numbers in plots indicate absolute numbers of QFL-Dex⁺ cells detected after enrichment. (**c**) Absolute numbers (left) and frequencies (right) of QFL-Dex⁺ cells detected (as in **b**) among TCRβ⁺CD4⁻ population in the spleen or siIEL compartment. **p=0.0069 (**d, f**) Analysis of Vα3.2 expression on QFL-Dex⁺(QFL) or total CD8⁺ T (Total) cells from the spleen (**d**) or the siIEL compartment (**f**). Numbers in plots indicate average percentages of Vα3.2⁺ cells within the indicated populations. (**e**) Frequencies of Vα3.2⁺ cells detected as in (**d**) ****p<0.0001 (**g**) Frequencies of Vα3.2⁺ cells detected as in (**f**) **p=0.0058 Representative data are shown in flow plots (**a, b, d, f**) and the number of replicates is specified in the bar graphs (**c, e, g**). Each symbol represents data collected from the indicated tissue isolated from an individual mouse. Data from ≧5 independent experiments were pooled for statistical analysis. p-values were calculated with student's t-test. 'ns' indicates the comparison was not significant.

The online version of this article includes the following figure supplement(s) for figure 1:

**Figure supplement 1.** Verification of QFL-Dextramers.

**Figure supplement 2.** QFL T cells are barely detectable in the TCRβ⁺CD4⁺ population.

ERAP1 deficient mice with the reduction being more significant in Qa-1ᵇ-KO mice. In contrast, the phenotype of splenic Vα3.2⁻ QFL T cells was not significantly affected (*Figure 3b and c*). Similar pattern of phenotype change was observed for the siIEL Vα3.2⁺ QFL T cells which showed almost complete loss of the unconventional natIEL phenotype including CD8αα expression in Qa-1ᵇ-KO mice (*Figure 3d and e*, *Figure 3—figure supplement 1*). These results demonstrate that TAP and Qa-1ᵇ both play a role in the establishment of the unconventional Vα3.2⁺ QFL T population, with TAP being required for the presence of the population, and Qa-1ᵇ being required for the unconventional pheno-type imprinting of these cells. Paradoxically, the loss of ERAP1, which would be expected to increase the presentation of FL9, also led to a reduction of memory and unconventional phenotypes in QFL

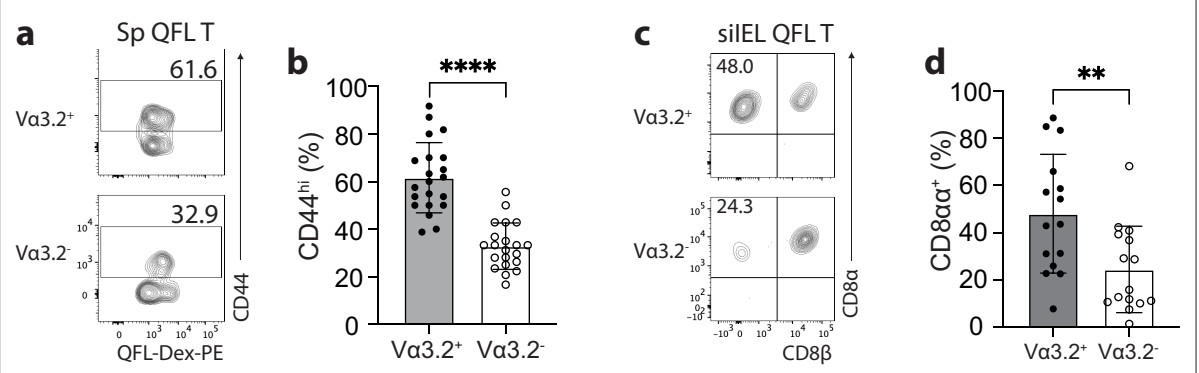

**Figure 2.** The unconventional phenotype of Vα3.2⁺ QFL T cells. (**a**) Analysis of CD44 expression on Vα3.2⁺ or Vα3.2⁻ QFL-Dex⁺ cells enriched from the spleen of naïve wild-type (WT) mice. Numbers in plots indicate average percentages of CD44ʰⁱ cells detected among the Vα3.2⁺ or Vα3.2⁻ QFL-Dex⁺ populations. (**b**) Frequencies of CD44ʰⁱ cells detected as in (**a**).****p<0.0001 (**c**) Analysis of CD8α and CD8β expression on Vα3.2⁺ or Vα3.2⁻ QFL-Dex⁺ cells enriched from the small intestinal intraepithelial lymphocyte (siIEL) compartment of naïve WT mice. Numbers in plots indicate average percentages of CD8αα⁺ cells detected among the Vα3.2⁺ or Vα3.2⁻ QFL-Dex⁺ populations. (**d**) Frequencies of CD8αα⁺ cells detected as in (**c**) **p=0.0065 Samples of <20 cells in the Vα3.2⁺QFL-Dex⁺ gate were excluded in the phenotype analysis. Representative data are shown in flow plots (**a, c**) and the number of replicates is specified in the bar graphs (**b, d**). Each symbol represents data collected from the indicated tissue isolated from an individual mouse. Data from ≧11 independent experiments were pooled for statistical analysis. p-values were calculated with student's t-test. 'ns' indicates the comparison was not significant.

The online version of this article includes the following figure supplement(s) for figure 2:

**Figure supplement 1.** Phenotype of total Vα3.2⁺ or Vα3.2⁻ T population in the spleen or small intestinal intraepithelial lymphocyte (siIEL) compartment.

T cells, perhaps due to the loss of certain QFL T clones as a result of tolerance mechanisms, such as negative selection. It is also worth noting that total splenic Vα3.2⁺CD8αβ⁺ T cells showed a similar loss of memory phenotype in Qa-1ᵇ-KO and ERAP1-KO mice, whereas Vα3.2⁻CD8αβ⁺T cells did not (*Figure 2—figure supplement 1c*). Thus, the unconventional Vα3.2⁺QFL T cells specifically, and a substantial proportion of total Vα3.2⁺ splenocytes, are impacted by Qa-1ᵇ and ERAP1.

## Vα3.2⁺QFL T cells are functionally innate-like in the spleen but are quiescent in the gut

To further investigate the functional features of Vα3.2⁺QFL T cells, we utilized transgenic mice (QFLTg) expressing the predominant invariant TCR found on QFL T cells (*Vα3.2Jα21, Vβ1Dβ1Jβ2–7*), referred to hereafter as QFLTg cells. Abundant QFLTg cells or control OT-1 cells were detected in both the spleen and siIEL compartment of the respective TCR transgenic mice (*Figure 4—figure supplement 1 a*). Phenotypically, the QFLTg cells recapitulated the unconventional phenotype of polyclonal Vα3.2⁺QFL T cells with high CD44 expression in the spleen and CD8αα expression in the siIEL compartment. In contrast, OT-1 cells were phenotypically similar to conventional naïve CD8⁺ T cells, as they expressed low to intermediate levels of CD44 in the spleen and lacked CD8αα expression in the gut (*Figure 4—figure supplement 1b, c*).

Cytokines including IL-15 and the combination of IL-7/IL-18 have been implicated in the maintenance and survival of both 'virtual' memory T cells and IELs (*Klose et al., 2014*; *Okazawa et al., 2004*; *Prasad et al., 2021*; *Schluns and Lefrançois, 2003*). We thus compared the proliferation of QFLTg and OT-1 cells in response to these cytokines. In keeping with their virtual memory surface marker phenotype, splenic QFLTg cells were significantly more responsive to IL-15 or IL-7/IL-18 stimulation than OT-1 cells (*Figure 4a*). At 72 hr, ~80% of QFLTg cells had proliferated in response to the cytokines, compared with only ~50% of OT-1 cells (*Figure 4b*). In contrast, T cells isolated from the siIEL compartment of QFLTg or OT-1 mice showed a distinct pattern of cytokine responsiveness from the splenic T cells. First, both QFLTg and OT-1 cells from siIEL were less responsive to IL-15 than the splenic population. Second, siIEL QFLTg cells showed significantly reduced proliferation in response to the cytokines, especially to the combination of IL-7/IL-18, as compared with OT-1 cells (*Figure 4c and d*).

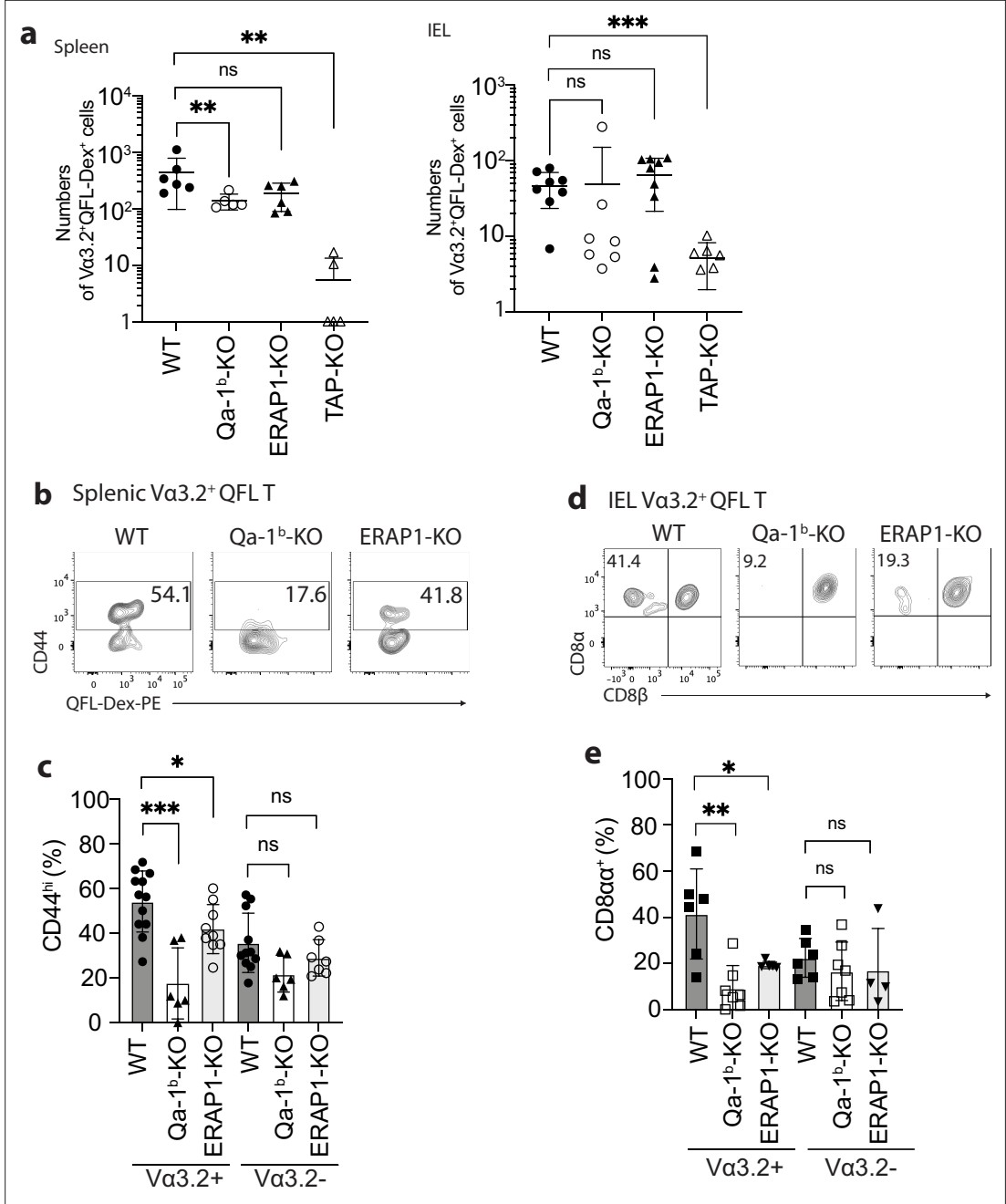

**Figure 3.** Phenotype of Vα3.2⁺ QFL T cells in mice of various genotypes. (**a**) Absolute numbers of Vα3.2⁺QFL-Dex⁺ cells detected in the spleen (left) or small intestinal intraepithelial lymphocyte (siIEL) compartment (right) of Qa-1ᵇ, ERAP1, or TAP deficient (Qa-1ᵇ-KO, ERAP1-KO, or TAP-KO) mice in comparison with wild-type (WT) mice.**p<0.009 ***p=0.0006 Symbols on x-axis indicate that the cells were undetectable in TAP-KO mice. (**b**) Analysis of CD44 expression on splenic Vα3.2⁺QFL-Dex⁺ cells enriched from naïve WT, Qa-1ᵇ-KO, or ERAP1-KO mice. Numbers in plots indicate average percentages of CD44ʰⁱ cells. (**c**) Frequencies of CD44ʰⁱ cells detected among Vα3.2⁺(as in **b**) or Vα3.2⁻ QFL T cells.***p=0.0004 *p=0.0388 (**d**) Flow cytometry analysis of CD8α and CD8β expression on the Vα3.2⁺QFL-Dex⁺ cells enriched from the siIEL compartment of naïve WT, Qa-1ᵇ-KO or ERAP1-KO mice. Numbers in plots indicate average percentages of CD8αα⁺ cells. (**e**) Percentages of CD8αα⁺ cells detected among Vα3.2⁺(as in **d**) or Vα3.2⁻ QFL T cells. **p=0.0082 *p=0.0335 Representative data are shown in flow plots (**b, d**) and the number of replicates is specified in the bar graphs (**a, c, e**). Each symbol represents data collected from the indicated tissue isolated from an individual mouse. Data from ≧4 independent experiments were pooled for statistical analysis. p-values were calculated with student's *t*-test. 'ns' indicates the comparison was not significant.

The online version of this article includes the following figure supplement(s) for figure 3:

**Figure supplement 1.** Qa-1ᵇ-dependent expression of natural IELs (natIEL) markers on Vα3.2⁺ QFL T cells.

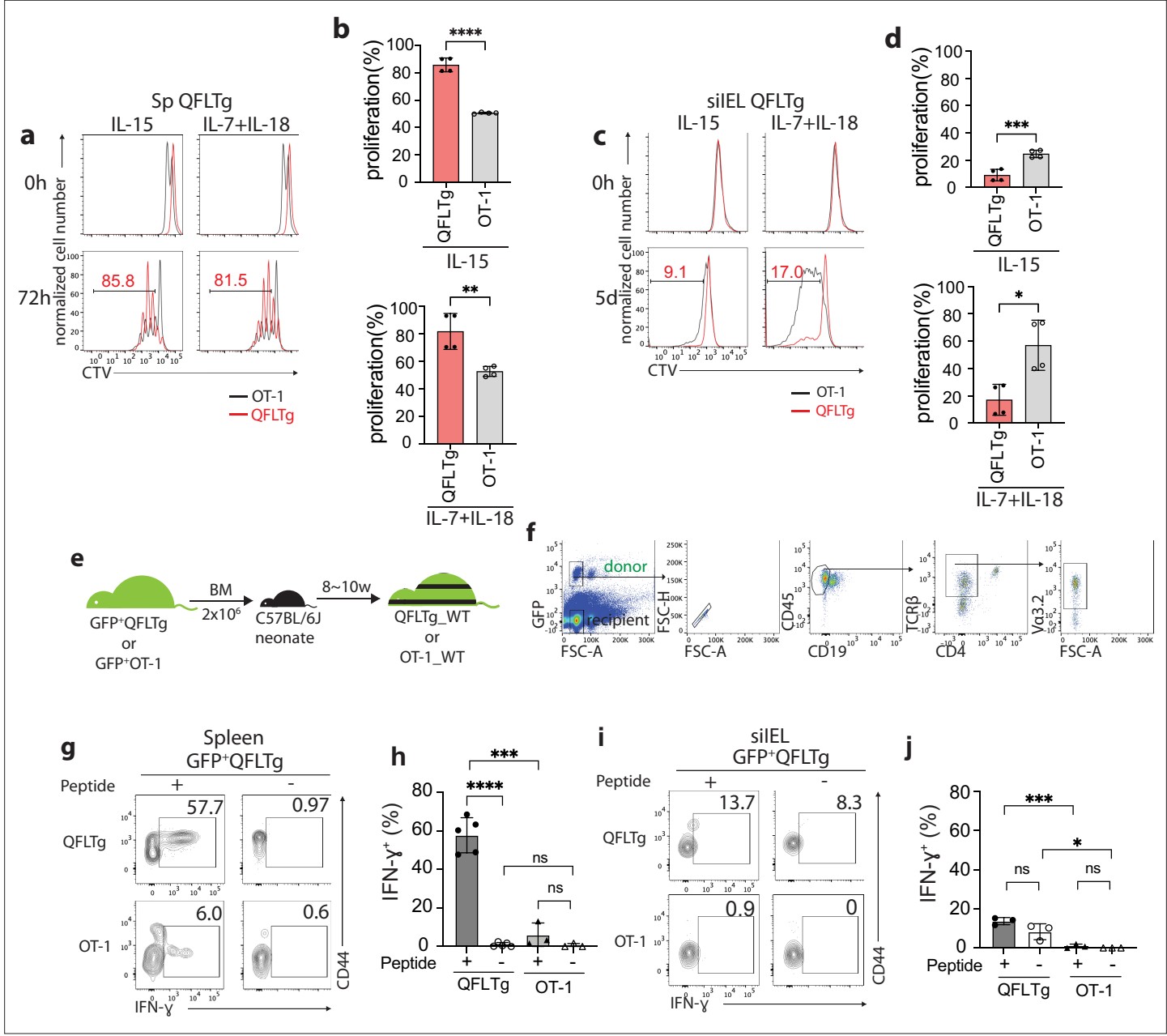

**Figure 4.** Distinct functional features of QFLTg cells in the spleen and gut. (**a, c**) Proliferation of splenic (**a**) or small intestinal intraepithelial lymphocyte (siIEL) (**c**) QFLTg or OT-1 cells from QFLTg or OT-1 mice in response to IL-15 (left) or a combination of IL-7 and IL-18 (right) stimulation. Cells were tracked using cell tracing violate (CTV). Numbers in plots indicate average percentages of proliferated QFLTg cells. (**b**) Percentages of proliferated splenic QFLTg or OT-1 cells as detected in (**a**) ****p<0.0001 **p=0.0052 (**d**) Percentages of proliferated siIEL QFLTg or OT-1 as detected in (**c**). ***p=0.0009 *p=0.0101 (**e**) Generation of QFLTg_WT or OT-1_WT partial hematopoietic chimera mice. 2 x 10⁶ of the bone marrow (BM) cells from GFP⁺QFLTg or GFP⁺OT-1 mice were transferred into wild-type (WT) neonates at 3~5 days of age. The chimera mice were analyzed at 8~10 weeks of age. (**f**) Gating strategy for the QFLTg cells originated from donor bone marrow cells in QFLTg_WT chimera mice. Plots representing donor-derived QFLTg population in the spleen being gated as GFP⁺CD45⁺CD19⁻TCRβ⁺CD4⁻Vα3.2⁺ cells. (**g, i**) Flow cytometry measurement of IFN-ɣ production by QFLTg or OT-1 cells isolated from the spleen (**g**) or siIEL compartment (**i**) of naïve chimera mice stimulated with or without 2 µM FL9 or SL8 peptide, respectively for 4.5 hr. Numbers in plots indicate average percentages of IFN-ɣ⁺ cells. (**h**) Percentages of IFN-ɣ⁺ cells detected as in (**g**) ****p<0.0001 ***p=0.0001 (**j**) Percentages of IFN-ɣ⁺ cells detected as in (**i**) ***p=0.0004 *p=0.0247 Representative data are shown in flow plots (**a, c, f, g, i**) and the number of replicates is specified in the bar graphs (**b, d, h, j**). Data from 2 (**b**), (**d**) or 3 (**h, j**) independent experiments were pooled for statistical analysis. Each symbol represents data collected from the indicated tissue isolated from an individual mouse. p-values were calculated with student's *t*-test. 'ns' indicates the comparison was not significant.

The online version of this article includes the following figure supplement(s) for figure 4:

**Figure supplement 1.** Phenotypes of QFLTg and OT-1 cells from the corresponding transgenic mice.

To ensure that altered effector function was not skewed by the high specific TCR frequency in TCR transgenics, we generated partial hematopoietic chimeric mice with physiological precursor frequencies (QFLTg_WT) by transferring bone marrow cells from GFP+QFLTg mice into nonirradiated naïve WT neonates (*Ladi et al., 2008*). OT-1_WT chimera mice were generated in parallel as representative of conventional CD8+ T cells (*Figure 4e*). This experimental design allowed us to track the GFP+ donor-derived T cells (GFP+CD45+CD19-TCRβ+CD4-Vα3.2+) separately from the recipient's endogenous T cells (*Figure 4f*). Based on the hyperresponsiveness of splenic QFLTg cells to cytokines, we hypothesized that these cells would exert effector functions more rapidly than conventional CD8 T cells, similar to the known innate-like T cells such as iNKT. We measured IFN-γ production by QFLTg or OT-1 cells from splenocytes or siIEL cells of naïve QFLTg_WT or OT-1_WT chimera mice stimulated with 2 µM FL9 or SIINFEKL(SL8) peptide, respectively for 4.5 hr directly ex vivo. A large fraction (~60%) of splenic QFLTg cells expressed high levels of IFN-γ under these conditions, while only 6% of OT-1 cells did so (*Figure 4g and h*). Notably, despite the low responsiveness to epitopes, siIEL QFLTg cells showed relatively higher basal levels of IFN-γ expression than OT-1 cells (*Figure 4i and j*). In contrast to the hyperresponsiveness of splenic QFLTg cells to their ligand, neither QFLTg nor OT-1 cells from the siIEL compartment showed significant IFN-γ production at 4.5 hr. The low IFN-γ production of IEL QFLTg cells was in keeping with their natIEL phenotype - antigen-experienced yet functionally quiescent (*Cheroutre et al., 2011*; *Denning et al., 2007*). We conclude that Vα3.2+QFL T cells in the spleen are functionally innate-like, whereas the population with the same TCR specificity in the small intestine shows functional features of natIEL.

## Gut microbiota is associated with retention but not homing of QFL T cells

The establishment and function of the intestinal immune system are intimately associated with the homeostasis of the gut microbial community, as evidenced by an altered gut immune cell composition in germ-free (GF) mice (*Belkaid and Hand, 2014*; *Macpherson and Harris, 2004*; *Round and Mazmanian, 2009*). To investigate if an association exists between gut microbiota and gut QFL T cells, we compared the numbers and frequencies of QFL T cells in the spleen and siIEL compartment of naïve specific-pathogen-free (SPF) and GF WT mice of various ages. First, we found the absolute number of QFL T cells in the spleen was unaffected by the absence of gut microbiota in either young or old mice. The frequency of QFL T cells was relatively higher in GF than in SPF mice, as a result of a decreased number of non-QFL CD8+ T cells in the spleen of GF mice (*Figure 5a and b*). However, the percentage of CD44hi cells within the total CD8+ T population was proportionally higher in the spleen of GF WT than in SPF WT mice regardless of their Vα3.2 expression (*Figure 5—figure supplement 1a*). This observation suggested that gut microbiota might be more closely associated with the presence of conventional naïve CD8+ T cells rather than the memory phenotype CD8+ T population in the spleen. Second, while the number and frequency of QFL T cells in siIEL compartment showed no significant difference between GF and SPF mice of relatively young age (8~17 week), the population was gradually lost in old GF mice (18~22 week) (*Figure 5c and d*), with the decrease in GF mice starting from around 16 weeks of age (data not shown). In addition, the QFL T cells detected in the spleen or siIEL compartment of young SPF or GF mice showed similar expressions of Vα3.2, CD44, and CD8αα (*Figure 5—figure supplement 2*). We conclude that although QFL T cells could be home to the gut in the absence of microbiota, microbe-derived antigens or signals are required for their maintenance there. Additionally, we observed that the average percentage of CD8αα+ cells among the total TCRβ+CD4- T population was proportionally larger in GF mice than SPF mice regardless of Vα3.2 expression, although the Vα3.2- population showed a more significant increase (*Figure 5—figure supplement 1b*). This was mainly a result of a decreased number of CD8αβ+ T cells in the siIEL of GF mice (data not shown). Nevertheless, here we showed that Qa-1b restricted T cells could home to the small intestine epithelium and display a natIEL phenotype independent of gut microbiota.

## QFL T cells cross-react with a microbial antigen

We next investigated how the QFL T population was retained in the gut of SPF WT mice, hypothesizing that the presence of specific antigens might play a role. QFL T cells specifically recognize the Qa-1b-FL9 complex which is uniquely presented on ERAP1 deficient cells. Because it is unlikely this ligand is constantly presented in gut epithelium under homeostatic conditions (*Figure 6—figure*

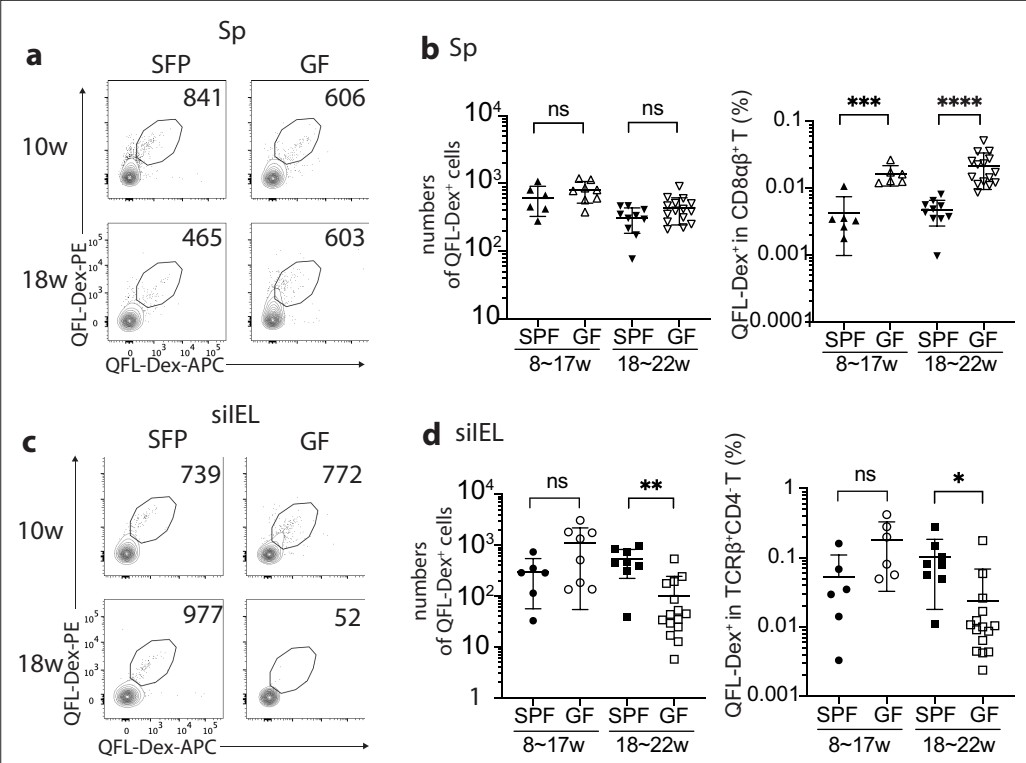

**Figure 5.** Gut microbiota is associated with retention but not homing of QFL T cells in the small intestinal intraepithelial lymphocyte (siIEL) compartment. (**a, c**) Flow cytometry of QFL-Dex⁺ cells enriched from the spleen (**a**) or siIEL compartment (**c**) of specific-pathogen-free (SPF) or germ-free (GF) wild-type (WT) mice at 10 or 18 weeks of age. Numbers in plots indicate absolute numbers of QFL-Dex⁺ cells. (**b**) Absolute numbers (left) and frequencies (right) of QFL-Dex⁺ cells detected within the total splenic CD8⁺ population in SPF or GF WT mice of 8~17 weeks or 18~22 weeks of age.***p=0.009 ****p<0.0001 (**d**) Absolute numbers and frequencies of QFL-Dex⁺ cells detected among the total siIEL TCRβ⁺CD4⁻ population in SPF or GF WT mice of 8~17 weeks or 18~22 weeks of age. **p=0.005 *p=0.037 Representative data are shown in flow plots (**a, c**) and the number of replicates is specified in the bar graphs (**b, d**). Each symbol represents data collected from the indicated tissue isolated from an individual mouse. Data from ≧5 independent experiments were pooled for statistical analysis. p-values were calculated with student's *t*-test. 'ns' indicates the comparison was not significant.

The online version of this article includes the following figure supplement(s) for figure 5:

**Figure supplement 1.** Presence of antigen experienced CD8⁺ T cells is independent of gut microbiota.

**Figure supplement 2.** QFL T cells detected in relatively young (8~17 week) specific-pathogen-free (SPF) or germ-free (GF) wild-type (WT) mice are phenotypically similar.

*supplement 1a*), we reasoned that QFL T cells might be retained in the gut through exposure to FL9 homolog peptide(s) expressed by commensal bacteria that colonize the small intestine.

To identify potentially cross-reactive peptides, we first determined the key residues of the FYAE-ATPML(FL9) peptide that could potentially affect the activation of QFL T cells. Due to the lack of structural information on QFL-TCR:FL9:Qa-1ᵇ ternary complex, we started by predicting the structure using AlphaFold2 (*Mayans et al., 2014*). The sequences of the α1 and α2 domain of Qa-1ᵇ α-chain, FL9, and Vα3.2Jα21, Vβ1Dβ1Jβ2–7 were submitted as input. Most of the residues in the predicted structure have high to intermediate confidence scores as was measured by the predicted local distance difference test (pLDDT). The output structure model of the top-ranked pLDDT score ('rank_1'=90.8) was selected for further visualization (*Figure 6—figure supplement 1b*). The simulated QFL-TCR:FL9:Qa-1ᵇ complex model suggests a similar TCR docking as observed in TCRαβ complexed with classical MHC I (*Figure 6a*). While we fully appreciated the limitations in accurately predicting TCR docking, the prediction of the pMHC complex is nevertheless relatively reliable based on the well-defined principles (*Nielsen et al., 2020*). Superimposition of simulated FL9:Qa-1ᵇ model

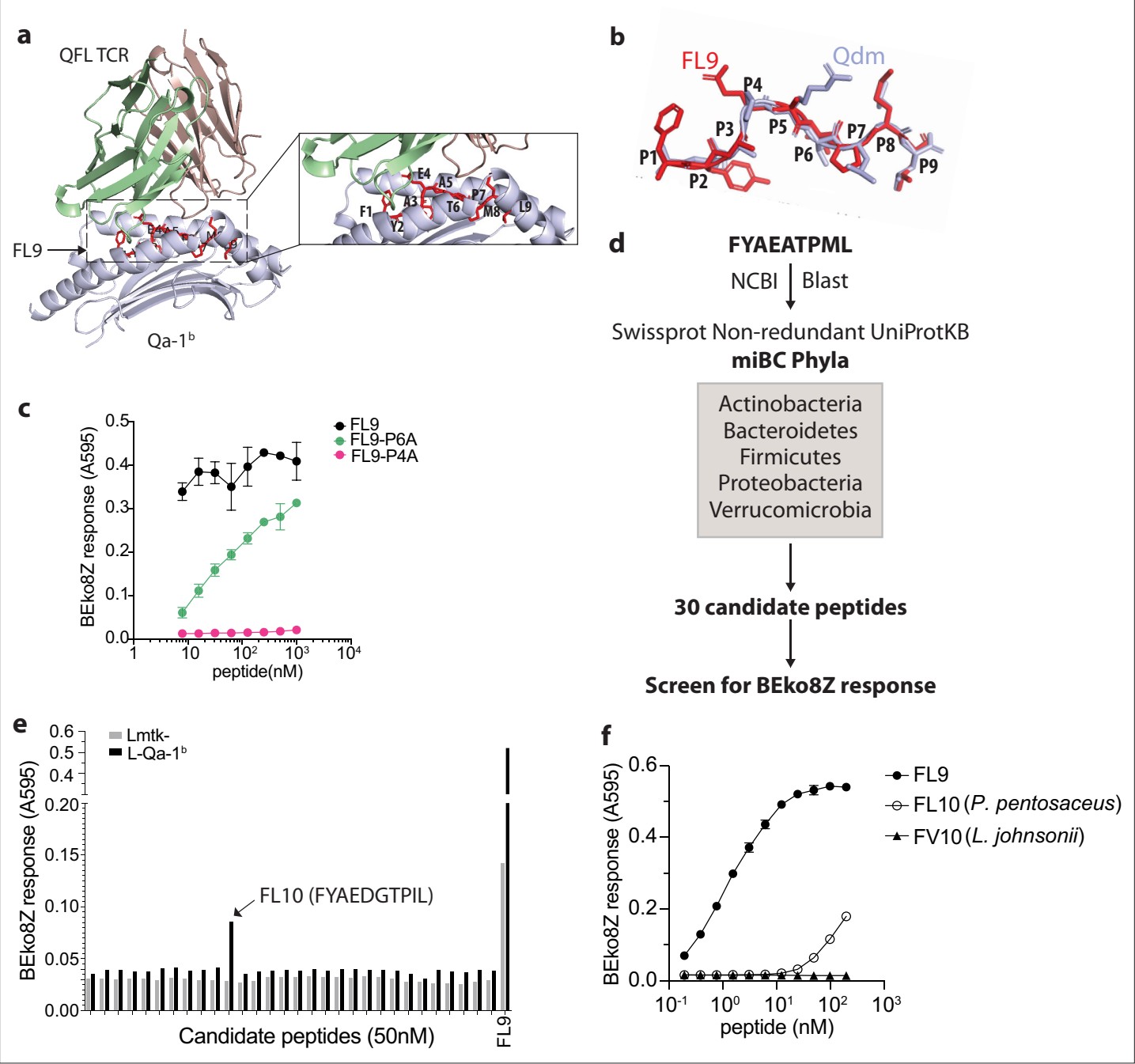

**Figure 6.** Identification of a QFL T cross-reactive FL9 homolog peptide expressed in commensal bacterium *P. pentosaceus*. (**a**) Structure model of the QFL TCR:FL9:Qa-1[b] complex predicted by AlphaFold2. Detailed view of the binding groove is shown in the right panel. (**b**) Conformation alignment of FL9 (red) and Qdm (periwinkle). (**c**) Alanine screen of the predicted key residues (**P4 and P6**) in FL9. Activation of QFL T cells was assessed by measuring the LacZ response of BEko8Z hybridoma cells to the Qa-1[b]-expressing Lmtk⁻ (L-Qa-1[b]) cells pulsed with mutant FL9 peptides including FYAEAAPML (FL9-P6A) and FYAAATPML(FL9-P4A) or the wild-type FYAEATPML(FL9) using spectrophotometry. (**d**) Workflow for identification of commensal bacterial FL9 homolog peptide which cross-activates QFL T cells. (**e**) Identification of the QFL T cross-reactive peptide FYAEDGTPIL(FL10). BEko8Z response to the L-Qa-1[b] cells or the non-Qa-1[b]-expressing Lmtk⁻ cells pulsed with 50 nM of the FL9 or the candidate homolog peptides. (**f**) Response of BEko8Z hybridoma to L-Qa-1[b] cells pulsed with FL9, FL10 or FYAEDDTPIV (FV10) peptide of various concentrations. (**c, e, f**) Data are from one experiment representative of three independent experiments.

The online version of this article includes the following figure supplement(s) for figure 6:

**Figure supplement 1.** Generation of a curated library of 30 candidate FL9 homolog peptides.

onto the crystallographic structure of Qdm:Qa-1$^b$ (PDB 3VJ6) suggested similar main chain conformations between FL9 and Qdm (*Figure 6b*). We thus refer to the reported structure of Qdm:Qa-1$^b$ whose anchoring residues were P2, P3, P6, P7, and P9 of AMAPRTLLL(Qdm) for determination of putative key residues on FL9 (*Zeng et al., 2012*). Among the rest of the non-anchoring residues, we hypothesized that the glutamic acid at P4 of FL9 may be critical for interacting with the QFL TCR. Notably, the importance of P4 for TCR interaction was also highlighted in recently published studies where crystal structures of several disease-related TCR:pMHC I complexes were reported including an αβTCR complexed with an HIV peptide loaded in HLA-B5301 (*Li et al., 2023*) and a pre-TCR complexed with vesicular stomatitis virus octapeptide(VSV8) loaded in H-2K$^b$ (*Li et al., 2021*). To test the hypothesis, we used the QFL T hybridoma cell line BEko8Z which expresses TCR-induced β-galactosidase (LacZ) to test whether the mutant FL9 peptide with the P4 residue substituted by alanine altered T cell responses. The P6 threonine whose side chain basically bends towards the binding groove with relatively limited access to TCR engagement was tested in parallel as a putative anchoring residue. By measuring the LacZ production of BEko8Z cells in response to Qa-1$^b$-expressing Lmtk$^-$ (L-Qa-1$^b$) cells pulsed with FYAAATPML (FL9-P4A) or FYAEAAPML (FL9-P6A), we found that in agreement with the prediction, replacement of the P6 threonine reduced TCR recognition, while replacement of the P4 glutamic acid led to the complete loss of the BEko8Z response (*Figure 6c*). Additionally, we performed Qa-1$^b$ binding predictions for various FL9 mutant peptides using NetMHCpan4.1 (*Reynisson et al., 2020*). The predicted binding scores suggested that substitution of the P9 Leu with Ala should decrease the binding of FL9 peptide to Qa-1$^b$, whereas mutation of the rest of the residues was less likely to have such an effect. Hence, Leu at P9 position may serve as a key anchoring residue (*Figure 6—figure supplement 1c*). Thus, P4, P6, and C-terminus positions were considered the key residues with the P4 Glu being particularly critical for QFL T recognition. Based on these data, the presence of glutamic acid at P4 was considered to be important, while the C-terminus leucine and P6 threonine were considered secondary. Although it is possible that the residues that were not tested in the T cell assays are involved in peptide binding or TCR interaction, data on the P4, P6, and P9 residues of FL9 peptide were sufficient for determining peptide homology and establishing a testable library of bacterial FL9 homologs.

To identify potential bacterial FL9 homologous peptides, we aligned the FL9 peptide sequence with proteomes of the five dominant gut commensal bacterial phyla, including Actinobacteria, Bacteroidetes, Firmicutes, Proteobacteria, Verrucomicrobia from the NCBI Swissprot Non-redundant UniProtKB database (*Donaldson et al., 2016*). The alignment result was further curated to 30 candidate peptides based on their homology with FL9 peptide as described above (*Figure 6d*, *Supplementary file 1*). By measuring LacZ production of BEko8Z cells in response to L-Qa-1$^b$ cells pulsed with 50 nM of each candidate peptide, we identified a single stimulatory peptide FYAEDGTPIL (FL10) that is expressed in *P. pentosaceus*, a gram-positive lactic acid bacteria (LAB) that colonizes in the small intestine (*Figure 6e*). While sensitivity to FL10 was ~1000 folds less than FL9 peptide, BEko8Z cells could recognize FL10 at concentrations as low as 1.5 nM. In contrast, FYAEDDTPIV(FV10), an FL10 homologous peptide expressed in *Lactobacillus johnsonii* which is a common LAB that colonizes the GI tract, failed to activate BEko8Z cells (*Figure 6f*, *Figure 6—figure supplement 1d*).

## QFL T cells are retained in the gut by a commensal bacterium

To further investigate the association between *P. pentosaceus* and Vα3.2$^+$QFL T cells in the gut, we first generated QFLTg_WT partial hematopoietic chimeras using germ-free WT mice (GF QFLTg_WT) and monocolonized the GF chimera mice with *P. pentosaceus* at 8 weeks of age. The mice were analyzed 8 weeks later (*Figure 7a*). The donor-derived QFLTg cells and the recipient-derived Vα3.2$^+$QFL T cells were distinguished by GFP expression and further defined as the CD45$^+$CD19$^-$TCRβ$^+$CD4$^-$Vα3.2$^+$QFL-Dex-PE$^+$ population on flow cytometry as was described earlier. While the donor-derived QFLTg populations were present and phenotypically unaltered in the spleen and siIEL compartment of GF QFLTg_WT chimera mice at 8 weeks of age, they were barely detectable in the GF chimera mice at 16 weeks of age (*Figure 7—figure supplement 1*, *Figure 7b*). Notably, monocolonization of the GF QFLTg_WT chimera mice with *P. pentosaceus* completely restored the donor-derived QFLTg population in both tissues in 16 week GF chimera mice (*Figure 7b*). We then further tested whether restoration of the gut QFL T population was specifically associated with *P. pentosaceus* by monocolonizing GF WT mice with *P. pentosaceus* or *L. johnsonii* (*Figure 7a*). Analysis of the microbiome composition

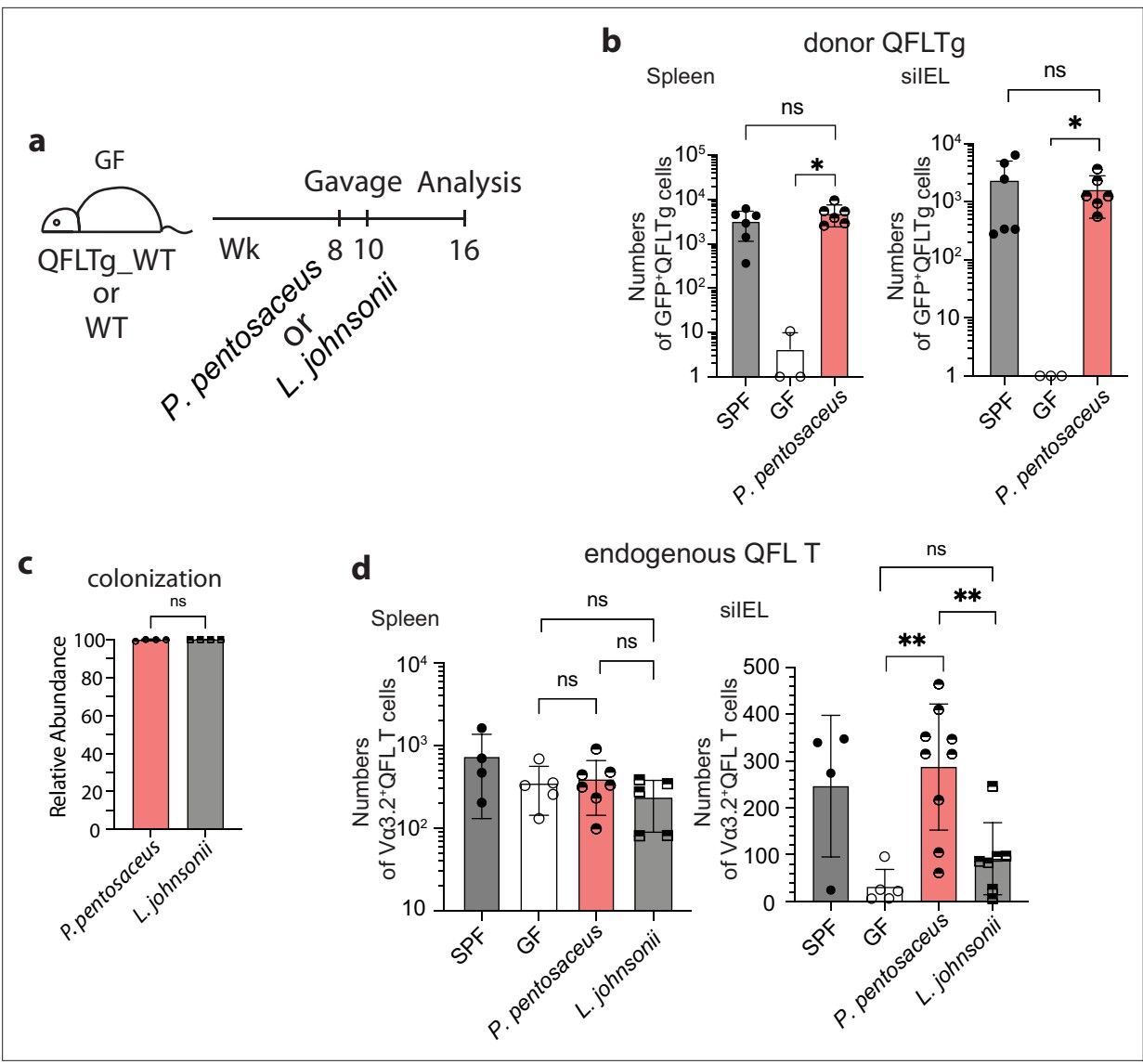

**Figure 7.** Association between Vα3.2+QFL T cells and the commensal bacterium *P. pentosaceus*. (**a**) Colonization of germ-free (GF) mice with commensal bacteria. GF QFLTg_WT chimera mice received oral gavage of *P. pentosaceus* at 8 weeks of age and were analyzed for both the donor-derived QFLTg cells and the endogenous Vα3.2+QFL T cells at 16 weeks of age. GF wild-type (WT) mice received oral gavage of *P. pentosaceus* or *L. johnsonii* at 8 and 10 weeks of age and were analyzed for the endogenous Vα3.2+QFL T cells at 16 weeks of age. (**b**) Absolute numbers of donor-derived QFLTg cells in the spleen (left) or small intestinal intraepithelial lymphocyte (siIEL) compartment (right) of 16 week GF QFLTg_WT chimera mice colonized with *P. pentosaceus* in comparison with specific-pathogen-free (SPF) or uncolonized GF chimera mice. Symbols on the x-axis indicate that QFLTg cells were undetectable in the indicated group. *p<0.03 (**c**) Relative abundance of *P. Pentosaceus* or *L. Johnsonii* was determined by whole shotgun metagenomic sequencing of microbiota composition in pooled fecal pallets collected from each cage. (**d**) Absolute numbers of endogenous Vα3.2+QFL T cells in the spleen (left) or siIEL compartment (right) of 16 week colonized GF mice compared with SPF or uncolonized GF chimera mice. Endogenous Vα3.2+QFL T data collected from the GF QFLTg_WT chimeric mice and the non-chimeric WT mice were pooled for this analysis. **p<0.008 The number of replicates is specified in the bar graphs with each symbol representing data collected from the indicated sample/tissue isolated from an individual cage (**c**) or mouse (**b, d**). Data from ≧3 independent experiments were pooled for statistical analysis. p-values were calculated with student's *t*-test. 'ns' indicates the comparison was not significant.

The online version of this article includes the following figure supplement(s) for figure 7:

**Figure supplement 1.** Donor-derived QFLTg cells are present and phenotypically similar in specific-pathogen-free (SPF) or germ-free (GF) QFLTg_WT chimera mice before 8 weeks of age.

**Figure supplement 2.** Age-associated loss of donor-derived QFLTg cells in the spleen of QFLTg_WT chimera mice.

**Figure supplement 3.** Unaltered phenotype of donor- or recipient-derived Vα3.2+ QFL T cells in 16 week germ-free (GF) chimera mice colonized with *P. pentosaceus*.

of fecal pallets showed that both species efficiently colonized the GF mice (*Figure 7c*). While the endogenous Vα3.2⁺QFL T population in the siIEL compartment was retained by *P. pentosaceus* colonization of GF mice at 16 weeks of age, *L. johnsonii* colonization failed to restore the population to a level comparable with SPF or *P. pentosaceus* colonized mice. On the other hand, maintenance of the endogenous splenic QFL T population was independent of microbes, in line with our earlier observations (*Figure 7d*). Donor-derived QFLTg and endogenous Vα3.2⁺QFL T cells in the spleen likely showed discrepant results due to different sources of precursors for the two populations. Unlike the endogenous populations which were constantly replenished, the donor-derived QFLTg cells rose from a fixed number of bone marrow cells, and gradually decreased in number as the animal age (*Figure 7—figure supplement 2*). Thus, the presence of the ligand which was supplied by a peptide derived from *P. pentosaceus* might be required for the donor-derived QFLTg cells to be retained in the spleen, whereas the endogenous splenic population was unaffected by microbes. Phenotype analysis revealed that both the donor-derived QFLTg cells and the endogenous Vα3.2⁺QFL T cells from the colonized GF chimera mice showed a memory phenotype identical to that seen in the SPF chimera mice (*Figure 7—figure supplement 3*). We thus conclude that the late-life maintenance of Vα3.2⁺QFL T cells in the small intestine was dependent on the commensal bacterium *P. pentosaceus* which expresses a FL9 homolog peptide. While the presence of the bacterial FL10 peptide likely plays an important role in the retention of gut QFL T cells, we do not exclude the possible contributions of other commensal bacterial factors. In addition, the unconventional phenotype of QFL T cells was likely acquired through microbe-independent mechanisms.

## Discussion

The small intestine epithelium harbors a highly heterogenous intraepithelial lymphocyte population which is comprised of a large proportion of CD4⁻CD8αβ⁻CD8αα⁺ T cells. Due to the heterogeneity, it has been challenging to identify or study a particular antigen-specific T cell clone from the population. Here, we found that QFL T cells, a Vα3.2-expressing CD8⁺ T cell population which specifically recognizes the Qa-1ᵇ-FL9 ligand presented on ERAP1-KO cells, naturally resided in both the spleen and small intestine epithelium of naïve WT mice. Further characterization of QFL T cells revealed their unconventional phenotype, functionality, and intimate association with gut microbiota (*Table 1*).

Unlike conventional memory T cells which are generated upon encounter with their cognate peptide antigens on classical MHC molecules, 'innate' memory T cells arise under homeostatic conditions in response to neonatal lymphopenia, high levels of IL-4 or exposure to self-antigens in naïve mice (*Jameson et al., 2015*; *Sprent and Surh, 2011*). Here, we found that splenic Vα3.2⁺QFL T cells display a memory phenotype and the corresponding rapid effector function, but acquire this phenotype in a Qa-1ᵇ dependent manner. Given the unique dependence on Qa-1ᵇ expression for the phenotype imprinting of QFL T cells, it is possible that QFL T cells are exposed to transiently induced QFL epitope on cells due to various intracellular stressors that might affect ERAP1 function. Alternatively, QFL T cells might cross-react with Qa-1ᵇ presented FL9 homolog peptide(s). Indeed, we found that Vα3.2⁺QFL T cells cross-reacted with one such variant of FL9 peptide (FL10) expressed in a commensal bacterium and presented by Qa-1ᵇ. However, the Qa-1ᵇ-FL10 ligand itself is unlikely to be directly associated with the memory imprinting for QFL T cells as these cells showed unaltered phenotypes in germ-free mice. There might be other unidentified self-peptide(s) presented by Qa-1ᵇ that are involved in the process. It is also possible that there is some level of degeneracy in the TCR-Qa-1ᵇ

**Table 1.** Characteristics of Vα3.2⁺ QFL T cells in the spleen and intestinal intraepithelial lymphocyte (IEL).
Summary of the characteristic features of the splenic and IEL Vα3.2⁺ QFL T cells including their cell type, expression of cell surface markers, dependence on microbiota for tissue homing/retention, and the critical molecules required for population establishment or phenotype imprinting.

| Tissue | Cell type | Phenotype | Microbiota | Critical molecules |
|---|---|---|---|---|
| Spleen | Innate-like T | CD44ʰⁱ | Independent | Population establishment: Qa-1ᵇ (partially required), TAP Phenotype imprinting: Qa-1ᵇ, |
| IEL | Natural IELs | CD8αα⁺ CD5⁻CD90⁻ | Homing: Independent Retention: Dependent | ERAP1 (partially required) |

interaction, so the precise peptide being presented is less critical than the presence of any peptide-Qa-1[b] complex. It is worth noting that the predicted structure of QFL-TCR:FL9:Qa-1[b] complex indicated that the FL9 peptide binds to Qa-1[b] in a manner potentially similar to Qdm. Because the binding groove of Qa-1[b] and its human homolog HLA-E display remarkable structural homology (*Zeng et al., 2012*), it is possible that HLA-E can present FL9 or its homolog peptides and further support the establishment of a QFL T-like CD8 T population. Studies on cytomegalovirus-vectored vaccines have shown that HLA-E-restricted CD8 T cells could recognize a broad repertoire of peptides presented upon vaccination and provide protection against HIV-1 infection (*Hansen et al., 2016*; *Yang et al., 2021*). Thus, analysis of the HLA-E peptide repertoire and investigation of the presence of HLA-E-restricted human version of QFL T cells will be promising and meaningful future directions.

In line with a previous study, we observed that the Vα3.2-expressing CD8αβ[+] T population was comprised of a relatively larger proportion of memory phenotype cells in the spleen than the non-Vα3.2 expressing population (*Prasad et al., 2021*). In addition, these memory phenotype non-QFL Vα3.2[+]CD8αβ[+] T cells were lost in Qa-1[b]-KO mice. These results indicate that Vα3.2QFL T cells are likely only one prototypical example of Qa-1[b] restricted Vα3.2[+]CD8[+] T cell clones with memory phenotype and innate-like function. We detected abundant Vα3.2[+]QFL T cells in small intestine epithelium which showed the signature phenotype of natIELs including expression of CD8αα and lack of CD4, CD8αβ, CD5, and CD90. Similar to the splenic population, while TAP is required for the presence of the population, the unconventional phenotype of Vα3.2[+]QFL T cells in the gut is strongly Qa-1[b]-dependent. It is worth noting that compared with the other Qa-1[b]-restrict T cells reported, such as a population of semi-invariant T cells which recognize TAP-independent peptides (*Doorduijn et al., 2018*) or an insulin peptide-specific CD8 T population (*Sullivan et al., 2002*), QFL T is likely a distinct population which undergoes alternative developmental stages which requires TAP-dependent presentation of FL9 (*Nagarajan et al., 2012*) (or other alternative essential peptides) for positive selection and Qa-1[b] for acquiring their unconventional features. Additionally, it is so far reported that CD8αα[+] IELs can be restricted to K[b]D[b], CD1d, or even unknown MHC Ib molecules (*Mayans et al., 2014*; *Ruscher et al., 2017*). Here by characterization of the gut Vα3.2[+]QFL T cells, we showed evidence of abundant Qa-1[b]-restricted TCRαβ[+] natIELs being present in the gut.

Functionally, the gut Vα3.2[+]QFL T cells showed features consistent with their CD8αα[+] natIEL phenotype. Notably, although the siIEL Vα3.2[+]QFL T cells showed delayed and reduced IFN-γ production in response to the cognate peptide, they displayed higher basal levels of intracellular IFN-γ than CD8αβ[+] IELs. It has been proposed that spontaneous IFN-γ secretion by IELs may be an important component of immunosurveillance at the mucosal surface, capable of identifying and eliminating transformed cells (*Carol et al., 1998*; *Reynisson et al., 2020*). Thus, siIEL QFL T cells may likewise be poised for immediate elimination of abnormal cells.

CD8αα[+] IELs are intrinsically programmed for innate functionality, as was shown by PMA/ionomycin-induced production of high levels of IFN-γ, CXCL2, etc. (*Van Kaer et al., 2014*). However, our observation of the relative hyporesponsiveness of siIEL Vα3.2[+]QFL T to stimuli indicated that under physiological conditions, the activation of IELs was more complex and highly regulated as TCRs and inhibitory coreceptors such as CD8αα were engaged (*Cheroutre and Lambolez, 2008*; *Macpherson and Harris, 2004*). It is not surprising that the gut QFL T cells were functionally quiescent as these cells are constantly challenged by the microbial or dietary antigens from the gut lumen (*Denning et al., 2007*). Cross-reaction between QFL T cells and peptide expressed in the commensal bacterium *P. pentosaceus* further supports this notion. Surprisingly, monocolonization of the GF mice with *P. pentosaceus* was sufficient to restore gut Vα3.2[+]QFL T cells that were lost in old GF mice. Of note, the control bacterium *L. johnsonni*, which lacks FL10 expression, did not restore gut Vα3.2[+]QFL T cells. Despite the evidence, we remain cautious of other possible signals generated from colonization of the commensal bacterium, such as secretion of bacteriocins.

In conclusion, Vα3.2-expressing QFL T cells represent an unconventional population of Qa-1[b]-restricted CD8[+] T cells which naturally reside at high frequency in small intestine epithelium. These cells have memory phenotypes in the spleen and reveal the characteristics of natIELs in the gut. In addition, the fact that Vα3.2[+]QFL T cross-reacts with FL9 homolog peptide expressed in a commensal bacterium suggests that these cells might be functionally versatile. Although we have not yet detected steady-state FL9 presentation, it remains to be assessed whether FL9 peptide can be transiently presented under certain conditions at particular locations or within certain time windows. Further

study of the biological significance of these cells could potentially reveal the delicate equilibrium between the gut microenvironment and the immune system.

## Methods

### Mice

SPF wild-type C57BL/6 J (000664), B6.129S6-H2-T23tm1Cant/J Qa-1b-KO (007907), C57BL/6-Tg (TcraTcrb)1100Mjb/J OT-1 (003831), or C57BL/6-Tg (UBC-GFP)30Scha/J B6.GFP (004353) mice were obtained from the Jackson Laboratory. QFL T TCR transgenic mice QFLTg were generated in the laboratory of E. Robey (University of California Berkeley, Berkeley, CA) and housed in our facility. GFP⁺Q-FLTg mice were generated in our facility by crossing B6.GFP mice with QFLTg mice. SPF QFLTg_WT, SPF OT-1_WT, and GF QFLTg_WT partial hematopoietic chimera mice were generated as previously described (*Ladi et al., 2008*). GF wild-type C57BL/6 J and GF QFLTg_WT chimera mice were generated and maintained in the Johns Hopkins Germ-free Mouse Core Facility. Mice were housed and all procedures were done in accordance with protocol (MO21M261) approved by the Animal Care and Use Committee of the Johns Hopkins University School of Medicine.

### Generation of the QFLTg mouse

The QFLTg TCR α-and β-chain sequences(*Tcra* and *Tcrb*) were cloned and amplified from the genomic DNA of BEko8Z hybridoma (*Guan et al., 2017*; *Nagarajan et al., 2012*). The *Tcra* was amplified with the forward primer 5′-AAAACCCGGGGCCAAGGCTCAGCCATGCTCCTGG-3′ and the reverse primer 5′- AAAAGCGGCCGCATACAACATTGGACAAGGATCCAAGCTAAAGAGAACTC-3′. The *Tcrb* was cloned with the forward primer 5′-AAAACTCGAGCCCGTCTGGAGCCTGATTCCA-3′ and the reverse primer 5′-AAAACCGCGGGGGGACCCAGGAATTTGGGTGGA-3′. The *Tcra* DNA fragment was cloned into the pTα cassette vector by inserting it between the XmaI and NotI sites, while the *Tcrb* DNA fragment was cloned into the pTβ cassette vector in between the XhoI and SacII sites (*Kouskoff et al., 1995*). The ampicillin resistance gene was removed from pTα and pTβ cassette by EarI enzyme digest. The QFLTg mice were generated on C57BL/6 J background in the Cancer Research Laboratory Gene Targeting Facility at UC Berkeley under standard procedures. Founder mice were identified by flow cytometry and PCR genotyping of tail genomic DNA using primers mentioned above.

### Antibodies, cell lines, and peptides

Antibody for flow cytometry were from BioLegend (anti-CD45(30-F11), anti-CD19(1D3/CD19), anti-TCRβ(H57-597), anti-CD8β(53–5.8), anti-Vα3.2(RR3-16), anti-Vα2(B20.1), anti-CD90.2 (53–2.1), anti-CD5(53–7.3), anti-IFN-γ(XMG1.2), anti-CD62L(MEL-14), and BD Biosciences anti-CD4(RM4-5), anti-CD8α (53–6.7), anti-CD44(IM7)). BEko8Z, L-Qa-1ᵇ, or Lmtk⁻ cells were maintained as previously described (*Nagarajan et al., 2012*). Peptides were obtained from GenScript. The purity of SL8, FL9, and FL10 peptides was ≥98%.

### Generation of the QFL-dextramer

Qa-1ᵇ-FL9 (QFL) monomers were synthesized by the Tetramer Core Facility of the US National Institutes of Health. Phycoerythrin (PE)- or allophycocyanin (APC)-conjugated streptavidin were obtained from Agilent. The QFL dextramers were generated following the 'dextran doping' technique (*Bethune et al., 2017*). The QFL monomers were incubated with PE or APC-conjugated streptavidin at a molar ratio of 3:1 at 4°C for 10 min followed by the addition of biotinylated dextran molecular weight 500 kDa at a molar ratio of 1:20 with respect to streptavidin. The mixture was further incubated at 4°C for ≥1 hr before being used in experiments.

### Isolation of IELs

Small intestinal IELs were isolated following an established protocol with minor modifications (*Qiu and Sheridan, 2018*). In brief, the small intestine with Payer's patches removed was cut into appropriate length. The fecal content and mucus were removed by expelling with the flat side of forceps followed by flushing with PBS. The intestine was cut open longitudinally to reveal the epithelium and further cut laterally into ~2 cm pieces. The intestine pieces were then placed in 25 ml warmed dithioerythritol (DTE) solution (Ca²⁺- and Mg²⁺-free Hanks balanced salt solution, HEPES-bicarbonate buffer, 10% FCS)

in 50 ml conical tube and were shaken at 75 rpm 37°C for 20 min. The tube was vortexed for 10 s before the supernatant was transferred into a new 50 ml conical tube through a 70 μM cell strainer. The cells were palleted and resuspended in 44% Percoll solution (44% Percoll in RPMI 1640). Density gradient was generated by underlaying the cell suspension with 67% Percoll solution (67% Percoll in RPMI 1640). The gradient cell suspension was further centrifuged at 1600 × g for 20 min at RT without using the brake. The layer of cells at the 44% and 67% interphase were thus collected as IELs.

## Enrichment for dextramer-positive cells

Splenic dextramer-positive cells were enriched as previously described (*Nagarajan et al., 2012*). IELs were resuspended in 100 μl sorter buffer (0.1% sodium azide and 5% FCS in PBS). PE or APC labeled QFL dextramers were added at a final dilution of 1:100. Cells were incubated at room temperature for 45 min, then washed twice with 3 ml of sorter buffer. 20 μl of anti-PE and 20 μl of anti-APC microbeads (Miltenyi Biotec) were added into cells resuspended in 450 μl of sorters buffer, followed by incubation of 20 min at 4°C. Cells were washed twice with 3 ml sorter buffer. The PE- and APC-labeled cells were positively selected by passing through LS magnetic columns (Miltenyi Biotec). The entire isolated population was stained with anti-CD45, anti-CD19, anti-TCRβ, anti-CD4, anti-CD8α and anti-CD8β. QFL T cells were gated as CD45+CD19-TCRβ+CD4-QFL-Dex-PE+QFL-Dex-APC+ population. The absolute numbers and frequencies of the cells were calculated based on a fixed number of CountBright Beads (Thermo Fisher) added into each sample.

## Cytokine stimulation and proliferation assay

Splenocytes or small intestinal IELs isolated from QFLTg or OT-1 mice were depleted of non-T cells using Pan T isolation Kit II (Miltenyi Biotec). The negatively selected T cells were then labeled with 5 μM Cell Tracing Violet (CTV) (Thermo Fisher) and adjusted to the concentration of $10^6$ /ml. $10^6$ of the CTV labeled T cells were stimulated in vitro for 3 or 5 days in cultures containing 100 ng/ml IL-15 (BioLegend) or a combination of 100 ng/ml IL-7 (BioLegend) and 100 ng/ml IL-18 (BioLegend) in 24-well plate at 37°C. Proliferation of cells was measured by dilution of CTV on the flow cytometer.

## CTL assay

Lymphocytes isolated from the spleen or siIEL compartment of naïve QFLTg_WT or OT-1_WT chimera mice were cultured in vitro with 2 μM of FL9 or SL8 peptide, respectively together with GolgiPlug (BD Biosciences) for 4.5 hr. WT splenocytes were supplemented into siIEL cultures as APCs. Cells were then washed and stained for surface markers. For measurement of IFN-γ production, cells were fixed and permeabilized using the Fixation/Permeabilization Kit (BD Biosciences) and stained for intracellular IFN-γ.

## Hybridoma assay

$10^5$ BEko8Z hybridoma cells were cocultured with $10^5$ L-Qa-1[b] or Lmtk- cells in a medium containing peptides of indicated concentrations in 96-well plates at 37°C for 12 hr. T cell response was determined by cleavage of the chromogenic LacZ substrate chlorophenol red-β-D-galactopyranoside (CPRG, Millipore Sigma) by TCR-induced LacZ and measured by spectrophotometry as presented by absorbance at 595 nm (A595) (*Sanderson and Shastri, 1994*). Background signals were subtracted by measurement of absorbance at 655 nm.

## Commensal culture and colonization

*Pediococcus pentosaceus Mees* (33314) and *Lactobacillus johnsonii* (33200) were obtained from ATCC. The bacteria were grown overnight at 37°C in 416 Lactobacilli MRS Broth (BD Biosciences). GF QFLTg_WT chimera mice were gavaged with 200 μl of 2 × $10^8$/ml CFU of *P. pentosaceus Mees* at 8 weeks of age. GF WT mice were gavaged with 200 μl of 2 × $10^8$/ml CFU of *P. pentosaceus Mees* or *L. johnsonii* at 8 and 10 weeks of age. Fecal pallets collected from the colonized mice were homogenized in 416 Lactobacilli MRS Broth, and the associated mice were determined by plating the appropriate dilution of homogenate on 416 Lactobacilli MRS agar plates. Bacterial colonization efficiency was determined by whole shotgun metagenomic sequencing of fecal pallets collected from the colonized mice at 16 weeks of age (TransnetYX Microbiome).

## AlphaFold2 structure and Qa-1$^b$ binding predictions

The predicted structure of QFL TCR:FL9:Qa-1$^b$ complex was obtained by submitting the sequences of the α1 and α2 domain of Qa-1$^b$ α-chain, FL9, and Vα3.2Jα21, Vβ1Dβ1Jβ2–7 as 'hetero-oligomer' to ColabFold v1.5.2-patch: AlphaFold2 using MMseqs2 at https://colab.research.google.com/github/sokrypton/ColabFold/blob/main/AlphaFold2.ipynb. The prediction was conducted using the alphafold2_multimer_v3 model on default settings. Visualization and alignment of the protein structures were conducted on the PyMOL software.

The Qa-1$^b$ binding prediction was conducted by submitting the sequences of Qdm, FL9, or mutant FL9 peptides and Qa-1$^b$ (UniProt: P06339) to NetMHCpan4.1 at https://services.healthtech.dtu.dk/services/NetMHCpan-4.1/.

## Acknowledgements

We thank Dr. X Li at the University of Science and Technology of China and the Ragon Institute of MGH, MIT, and Harvard for providing insights on the prediction of peptide binding to unconventional MHC I molecules, Dr. L Coscoy at the University of California, Berkeley for providing cellular biology expertise, and Mr. P Rhee at Shastri Lab, and Dr. H Ding at the JHU germ-free core facility for technical support. This work was supported by grants R01AI130210, R37AI060040, R01AI149341 and 1R01AI136972. Grants are funded by the National Institutes of Health (NIH), the National Institute of Allergy and Infectious Diseases (NIAID) to Dr. Nilabh Shastri and transferred to SS-N upon his passing. ICMJE guidelines have been followed.

## Additional information

### Funding

| Funder | Grant reference number | Author |
| --- | --- | --- |
| National Institutes of Health | R01AI130210 | Jian Guan<br>J David Peske<br>Michael Manoharan Valerio<br>Chansu Park<br>Ellen A Robey<br>Scheherazade Sadegh-Nasseri |
| National Institutes of Health | R37AI060040 | Jian Guan<br>J David Peske<br>Michael Manoharan Valerio<br>Chansu Park<br>Ellen A Robey<br>Scheherazade Sadegh-Nasseri |
| National Institutes of Health | R01AI149341 | Jian Guan<br>J David Peske<br>Michael Manoharan Valerio<br>Chansu Park<br>Ellen A Robey<br>Scheherazade Sadegh-Nasseri |

The funders had no role in study design, data collection and interpretation, or the decision to submit the work for publication.

### Author contributions

Jian Guan, Conceptualization, Resources, Data curation, Software, Formal analysis, Supervision, Validation, Investigation, Visualization, Methodology, Writing – original draft, Project administration, Writing – review and editing; J David Peske, Conceptualization, Writing – original draft, Writing – review and editing; Michael Manoharan Valerio, Resources, Validation, Methodology; Chansu Park, Resources, Methodology; Ellen A Robey, Conceptualization, Resources, Supervision, Methodology,

Writing – review and editing; Scheherazade Sadegh-Nasseri, Conceptualization, Resources, Supervision, Funding acquisition, Project administration, Writing – review and editing

### Author ORCIDs
Jian Guan ⬩ https://orcid.org/0000-0002-0118-6578
Ellen A Robey ⬩ https://orcid.org/0000-0002-3630-5266
Scheherazade Sadegh-Nasseri ⬩ https://orcid.org/0000-0002-8127-1720

### Ethics
Mice were housed and all procedures were done in accordance with protocols approved by Animal Care and Use Committee of the Johns Hopkins University School of Medicine.

Reviewer #1 (Public Review): https://doi.org/10.7554/eLife.90466.3.sa1
Reviewer #2 (Public Review): https://doi.org/10.7554/eLife.90466.3.sa2
Reviewer #3 (Public Review): https://doi.org/10.7554/eLife.90466.3.sa3
Author Response https://doi.org/10.7554/eLife.90466.3.sa4

## Additional files

### Supplementary files
• Supplementary file 1. Library of the 30 candidate FL9 homolog peptides expressed in commensal bacteria.

• MDAR checklist

### Data availability
Raw data obtained through flow cytometry for calculation QFL T numbers are deposited at Dryad (https://doi.org/10.5061/dryad.vq83bk40q).

The following dataset was generated:

| Author(s) | Year | Dataset title | Dataset URL | Database and Identifier |
|---|---|---|---|---|
| Guan J, Peske JD, Manoharan Valerio M, Park C, Robey EA, Sadegh-Nasseri S, Shastri N | 2023 | Commensal Bacteria Maintain a Qa-1b-restricted Unconventional CD8+ T Population in Gut Epithelium | https://doi.org/ 10.5061/dryad. vq83bk40q | Dryad Digital Repository, 10.5061/dryad.vq83bk40q |

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
