## [Editor Report · eLife assessment]

This is an **important** study that investigates the role of commensal microbes and molecules in the antigen presentation pathway affecting the development and phenotype of an unusual population of T lymphocytes. The authors provide **compelling** evidence to identify a population of unconventional T cells that exist in the small intestinal epithelium, which appear to depend on commensal microbes, and show that a single commensal microbe (that encodes an antigen capable of weakly stimulating these cells) is sufficient to maintain this T cell population.

---

## [Referee Report · Reviewer #1 (Public Review)]

Guan et al. explored the mechanisms responsible for the development, maintenance, and functional properties of a specific subset of unconventional T cells expressing a Va3.2 T cell receptor that recognizes a peptide, QFL, presented by the class Ib protein Qa-1. Prior studies from this group showed that cells from mice deficient in the ER protease ERAAP elicit responses in wild-type animals enriched for Qa-1-restricted CD8 T cells. They further showed that a significant proportion of these responses were directed against the QFL peptide derived from a conserved protein with incompletely understood functions. Many of these so-called QFL T cells expressed Va3.2-Ja21, were present in the spleen of wild-type mice, and exhibited a memory-like phenotype. Due to their relatively low frequency and weak staining with Qa-1 tetramers, analyzing QFL T cells has been challenging. Therefore, the authors generated dextramers, which permitted them to more rigorously identify these cells. They confirmed some of their previous findings and further showed that Va3.2+ and Va3.2- QFL T cells were present in the intestinal epithelium, where they also express CD8alpha homodimers, a characteristic of most small intestinal intraepithelial lymphocytes (siIELs), and most similar to the so-called natural siIELs that acquire their innate functions in the thymus. The authors show that TAP but not Qa-1 or ERAAP expression are required for the development of these cells, and both Qa-1 and ERAAP are required for the natural siIEL phenotype. Some of these findings were confirmed using a new TCR transgenic mouse expressing the QFL TCR. They further show that retention but not homing of QFL T cells to the intestinal epithelium involves commensal microorganisms, and using in silico approaches, they identify a commensal that contains a peptide similar to QFL that can activate QFL T cells. Finally, they show that this organism, P. pentosaceus, can promote gut retention of QFL T cells when it is introduced into germ-free mice. From these findings, the authors conclude that the microbiota influence the maintenance of Qa-1-restricted T cells.

Comments:

1. The authors employ a number of new reagents and elegant approaches to explore the development, maintenance and functional properties of QFL T cells.

2. Generally, conclusions made are well supported by the data presented.

3. One limitation of the work is that the immunological functions of QFL T cells remain unclear.

4. In their revised manuscript, the authors present additional data that have appropriately addressed the reviewer comments.

---

## [Referee Report · Reviewer #2 (Public Review)]

Summary: CD8+ QFL T cells recognize a peptide, FYAEATPML (FL9), presented on Erap1-deficient cells. QFL T cells are present at a high frequency in the spleen of naïve mice. They express an antigen-experienced phenotype, and about 80% express an invariant TCRα chain Vα3.2Jα21.

Here, Guan and coll. report that QFL T cells are present not only in the spleen but also in the intestinal epithelium, where they display several phenotypic and functional peculiarities. The establishment of spleen and gut Vα3.2+ QFL T cells is TAP-dependent, and their phenotype is regulated by the presence/absence of Qa-1b and Erap1. Maintenance of gut Vα3.2+ QFL T cells depends on the gut microbiota and is associated with colonization by Pediococcus pentosaceus.

Strengths:

This article contains in-depth studies of a peculiar and interesting subset of unconventional CD8 T cells, based partly on generating two novel TCR-transgenic models.

The authors discovered a clear relation between the gut microbiome and the maintenance of gut QFL T cells. One notable observation is that monocolonization of the gut with Pediococcus pentosaceus is sufficient to sustain gut QFL T cells.

Weaknesses:

In the absence of immunopeptidomic analyses, the presence or absence of the FL9 peptide on various cell types is inferred based on indirect evidence. Hence, whether the FL9 peptide is presented by some cells that express Qa-1b but not Erap1 remains unknown.

Analyses of the homology between the FL9 and bacterial peptides were limited to two amino acid residues (P4 and P6). This limitation is mitigated in part by the justifications provided by the authors in the revised preprint.

The potential function of QFL T cells remains elusive. The present article should provide an incentive for further functional studies.

---

## [Referee Report · Reviewer #3 (Public Review)]

The authors investigate the role of commensal microbes and molecules in the antigen presentation pathway in the development and phenotype of CD8 T cells specific for the Qa-1b-restricted peptide FL9 (QFL). The studies track both endogenous QFL-specific T cells and utilize a recently generated TCR transgenic model. The authors confirm that QFL-specific T cells in the spleen and small intestine intraepithelial lymphocyte (IEL) pool show an antigen-experienced phenotype as well as unique phenotypic and innate-like functional traits, especially among CD8+ T cells expressing Va3.2+ TCRs. They find that deficiency in the TAP transporter leads to almost complete loss of QFL-specific T cells but that loss of either Qa1 or the ERAAP aminopeptidase does not impact QFL+ T cell numbers but does cause them to maintain a more conventional, naïve-like phenotype. In germ-free (GF) mice, the QFL-specific T cells are present at similar numbers and with a similar phenotype to SPF animals, but in older animals (>18w) there is a notable loss of IEL QFL-specific cells. This drop can be avoided by neonatal colonization of GF mice with the commensal microbe Pediococcus pentosaceus but not a different commensal, Lactobacillus johnsonii, and the authors show that P. pentosaceus encodes a peptide that weakly stimulates QFL-specific T cells, while the homologous peptide from L. johnsonii does not stimulate such cells.

This study provides new insights into the way in which the differentiation, phenotype, and function of CD8+ T cells specific for Qa-1b/FL9 is regulated by peptide processing and Qa1 expression, and by interactions with the microbiota. The approaches are well designed, the data compelling, and the interpretation, for the most part, appropriate.

The response to several of my concerns involved reference to a different manuscript from the authors (which has not been through peer review), and for point #3, it would have been useful to provide experimental evidence (e.g., competitive inhibition assays) to justify their hypothesis that P4 serves as a TCR contact while P6 may be a Qa-1b contact residue. Nevertheless, the authors have made considerable efforts to clarify their approaches and interpretation, which strengthens the manuscript.

---

## [Author Response]

The following is the authors’ response to the original reviews.

We appreciate the reviewers’ detailed corrections and insightful comments. We have revised our manuscript per reviewers’ recommendations by including new data and clarifications/expansion of the discussion on our findings. Please see below for details.

**Reviewer #1 (Recommendations For The Authors):**
1. The introduction notes that CD1d KO mice show reduced levels of Va3.2 T cells (Ruscher et al.), which is interesting because innate memory T cell development in the thymus often requires IL-4 production by NKT cells. Have the authors explored QFL T cells in CD1d KO and/or IL-4 KO mice? Since their QFL TCR Tg mice still develop QFL T cells (and these animals likely have very few thymic NKT cells), NKT cells may not be required for the intrathymic development of QFL T cells?

Answer: We agree that investigation on the role of NKT cells or IL-4 in QFL T cell development will greatly further our understanding of these cells.

We validated the finding that expression of the QFL TCR transgene largely repressed the expression of endogenous TCRα, as indicated by the low levels of endogenous Vα2 on mature CD8SP T cells in both thymus and spleen. However, the frequencies of Vα2 usage in CD4 SP thymocytes and splenocytes from QFL transgenic mice were similar to non-transgenic mice, confirming that they underwent positive selection using endogenous TCR rather than the QFL TCR. We thus do not exclude the possible presence of NKT cells in QFLTg mouse and their potential involvement in the QFL T cells development. Our manuscript here is mainly focused on investigating the peripheral phenotype of QFL T cells and their association with the gut microbiota environment. Investigations into the role of CD1d/IL-4 will be best addressed in our future studies.

2. The finding that Qa-1 expression is not required for the development of QFL T cells raises questions about other MHC products that may be involved. In this context, it is interesting that TAP-deficient mice develop few QFL T cells, for reasons that are unclear, but the authors may speculate a bit. In this context, it may be helpful for the authors to note whether TAP is required for QFL presentation to QFL T cells. Since Qa-1 is not required, and CD1d is still expressed in TAP KO mice, what then could be responsible for their defect in QFL T cell development?

Answer: This is a great point. Figure 2 (from (Valerio et al., 2023) on the development of QFL T cells) tested whether QFL TCR cross-react with other MHC I molecules.

We assessed the activation of pre-selection QFLTg thymocytes in response to various MHC I deficient DC2.4 cell lines. While the QFL thymocytes showed partially reduced activation when stimulated with Qa-1b deficient APCs, triple knock-out (KO) of Qa-1b, Kb, and Db in DC2.4 cells reduced activation close to background levels. However, double knock-out of Qa-1b with either Kb, or Db led to stimulation that was intermediate between the triple KO and Qa-1b-KO cell lines. These data suggest that Kb and Db may contribute to the positive selection of QFL T cells in Qa-1b-KO mice.

TAP is required for FL9 peptide presentation and is very likely needed for presentation of the yet unidentified MHC Ia presented peptide(s) that are essential to QFL T positive selection. While CD1d/NKT cells/IL-4 may be involved in supporting the maturation of QFL T cells, we think in the TAP-KO mice the absence of TAP led to deletion/altered selection of the QFL T population at early developmental stage. We have added clarification on this point in the revised manuscript (line 412~418).

3. It may be worthwhile for the authors to note that Qa-1 was also dispensable for the intrathymic selection of another Qa-1-restricted TCR (Doorduijn et al. 2018. Frontiers Immunol.), although this is presumably not the case for others (Sullivan et al. 2002. Immunity 17, 95).

Answer: We appreciate this recommendation. We have noted this point in the resubmitted manuscript (line 412~418).

4. Lines 122-124: The sentence "Interesting ..." seemed confusing to me; are the numbers (60 and 30%) correct?

Answer: The numbers 60% and 30% were referring to the largest number we have detected for percentages of Va3.2 QFL T cells and Va3.2 CD8 T cell respectively. Here in the revised version, we replaced these numbers with average percentages (20.1% and <10%) to avoid confusion (line 134).

5. Qa-1/peptide complexes may also be recognized by CD94/NKG2 receptors, which may complicate the interpretation of the data (e.g., staining of the dextramers). From their previous work, it appears that Qa-1/QFL does not bind CD94/NKG2, which would be helpful to note in the text.

Answer: We have noted this point in the revised manuscript (line 117~121).

6. It would be helpful to add a few comments about the potential relevance to HLA-E.

Answer: We have included discussion on this point (line 391~401).

7. Figure legends: Most legends note the total number of replicates, which is usually quite high. It would also be helpful to indicate the total number of independent experiments performed and, when relevant, that the data are pooled from multiple independent experiments.

Answer: Thank you for raising the concern. We have clarified the experimental repeats in figure legends.

**Reviewer #2 (Recommendations For The Authors):**
1. The work of Nilabh Shastri was the foundation of the present study. Unfortunately, he passed away in 2021. Since he can no longer assume the responsibilities of a senior author, I wonder if it would be more appropriate to dedicate this paper to him than to list him as a co-author.

Answer: We have removed Dr. Shastri’s name as a co-senior author and have dedicated this work to his memory.

2. The official symbol for ERAAP is Erap1.

Answer: We have replaced ERAAP with ERAP1.

3. Please refrain from editorializing. For example, "strikingly" appears eight times and "interestingly" 9 times in the manuscript. Most readers believe they do not need to be said when something is striking or interesting.

Answer: We appreciate the Reviewer’s suggestion and have removed ‘strikingly’ and ‘interestingly’ from the manuscript.

4. In WT mice, are there some cell types that express Qa-1b but not Erap1 and could therefore present the FL9 peptide?

Answer: This is a great question. Using our highly sensitive QFL T cell hybridoma line BEko8Z(sensitivity shown in Fig. 6b), we have so far not been able to detect steady-state FL9 presentation by cells isolated from the spleen, lymph nodes, various gut associated lymphoid tissues or intestinal epithelial cells (Supplementary Fig. 8 a left panel). However, we do not exclude the possibility of FL9 peptide being transiently presented under certain conditions (i.e. ER stress/transformed cells) at particular locations or within certain time windows, which is of great importance for understanding the function of these cells but is beyond the scope of this study.

5. Since you have not tested substitutions at other positions, could you explain your reasoning that P4 and P6 are the critical residues (lines 271-272)?

Answer: Thank you for raising the concern. We have expanded on explanation of our strategy for determining peptide homology (line 272~313) in the revised manuscript. We have also included data on the structure the QFL TCR: FL9-Qa-1b complex predicted by Alphafold2, conformation alignment of FL9 and Qdm (Figure 6. a, b) and the NetMHCpan prediction of Qa1b binding of Qdm, FL9 and various FL9 mutant peptides (Supplementary Fig. 8 c) to help readers visualize the reasoning behind our strategy.

6. Readers might appreciate having a Figure summarizing the differences between spleen and gut QFL T cells.

Answer: This is a great suggestion. We have added a table summarizing the characteristic features of the splenic and IEL QFL T cells (Table 1).

7. In the discussion, readers would like to know what plan you might have to elucidate the function of QFL T cells.

Answer: We appreciate the recommendation. We have elaborated on our opinions and future directions in the resubmitted manuscript (line 393~401, 446~455).

**Reviewer #3 (Public Review):**
1. For most of the report, the authors use a set of phenotypic traits to highlight the unique features of QFL-specific CD8+ T cells - specifically, CD44high, CD8aa+ve, CD8ab-ve. In Supp. Fig. 4, however, completely distinct phenotypic characteristics are presented, indicating that IEL QFL-specific T cells are CD5low, Thy-1low. No explanation is provided in the text about whether this is a previously reported phenotype, whether any elements of this phenotype are shared with splenic QFL T cells, what significance the authors ascribe to this phenotype (and to the fact that Qa1-deficiency leads to a more conventional Thy-1+ve, CD5+ve phenotype), and whether this altered phenotype is also seen in ERAAP-deficient mice. At least some explanation for this abrupt shift in focus and integration with prior published work is needed. On a related note, CD5 expression is measured in splenic QFL-specific CD8+ T cells from GF vs SPF mice (Supp. Fig. 9), to indicate that there is no phenotypic impact in the GF mice - but from Supp. Fig. 4, it would seem more appropriate to report CD5 expression in QFL-specific cells from the IEL, not the spleen.

Answer: Expression of CD8αα and lack of CD4, CD8αβ, CD5 and CD90 expression was indeed reported as the characteristic phenotype of natIELs. We have clarified this point in the resubmitted manuscript (line 80). The CD8αα+ IEL QFL T cells have consistently showed CD5CD90- phenotype. While CD8αα expression was sufficient to describe their natIEL phenotype, we showed the CD5-CD90- data in Supplementary figures only to provide additional evidence.

The CD5 molecule by itself reflects the TCR signaling strength and high CD5 level is associated with self-reactivity of T cells (Azzam et al., 2001; Fulton et al., 2015). The implication of CD5 expression on QFLTg cells is discussed in our other manuscript where we investigate the development of these cells (Valerio et al., 2023). In Supplementary Fig. 9, because the donor splenic QFLTg cell have consistently showed comparable CD5 level between the GF and SPF group, we reasoned that it would not interfere with our interpretation of the CD44 expression.

2. The authors suggest the finding that QFL-specific cells from ERAAP-deficient mice have a more "conventional" phenotype indicates some form of negative selection of high-affinity clones (this result being somewhat unexpected since ERAAP loss was previously shown to increase the presentation of Qa-1b loaded with FL9, confirmed in this report). It is not clear how this argument aligns with the data presented, however, since the authors convincingly show no significant reduction in the number of QFL-specific cells in ERAAP-knockout mice (Fig. 3a), and their own data (e.g. Fig. 2a) do not suggest that CD44 expression correlates with QFL-multimer staining (as a surrogate for TCR affinity/avidity). Is there some experimental basis for suggesting that ERAAP-deficient lacks a subset of high affinity QFL-specific cells?

Answer: We think the presence of QFL T cells in ERAAP-KO mice is a result of the unconventional developmental mechanism of these cells which is better addressed in our complementary manuscript on the development of QFL T cells(Valerio et al., 2023). Valerio et al. found that the most predominant QFL T clone which expresses Vα3.2Jα21, Vβ1Dβ1Jβ2-7 received relatively strong TCR signaling and underwent agonist selection during thymic development, indicating that the QFL ligand is involved in selection of the innate-like QFL T population.

We agree that there is so far no direct evidence showing the QFL T cells that were absent in the ERAAP-KO mice were high-affinity clones. We have removed ‘high-affinity’ from the manuscript (line 180). While CD44 expression has been associated the antigen-experiences phenotype of T cells, it is yet unclear whether expression level of this molecule directly reflects TCR affinity/avidity. identification of clones of different affinities/avidities require high precision technologies that are not currently available to the research community. While we do have zMovi, a newly developed (developing) technology, in the lab claimed to measure relative avidity/affinity of different cell types for ligands, during the past two years working with this instrument has taught us that the technology is not yet advanced enough; it can only produce reliable data on extreme differences of single clones, i.e., high numbers of homogeneous cell types expressing very high affinity receptors.

3. The rationale for designing FL9 mutants, and for using these data to screen the proteomes of various commensal bacteria needs further explanation. The authors propose P4 and P6 of FL9 are likely to be "critical" but do not explain whether they predict these to be TCR or Qa-1b contact sites. Published data (e.g., PMID: 10974028) suggest that multiple residues contribute to Qa-1b binding, so while the authors find that P4A completely lost the ability to stimulate a QFL-specific hybridoma, it is unclear whether this is due to the loss of a TCR- or a Qa-1-contact site (or, possibly, both). This could easily be tested - e.g., by determining whether P4A can act as a competitive inhibitor for FL9-induced stimulation of BEko8Z (and, ideally, other Qa-1b-restricted cells, specific for distinct peptides). Without such information, it is unclear exactly what is being selected in the authors' screening strategy of commensal bacterial proteomes. This, of course, does not lessen the importance of finding the peptide from P. pentosaceus that can (albeit weakly) stimulate QFL-specific cells, and the finding that association with this microbe can sustain IEL QFL cells.

Answer: Thank you for raising the concern. We have expanded on explanation of our strategy for determining peptide homology (line 272~313) in the revised manuscript. We have also included data on the structure the QFL TCR: FL9-Qa-1b complex predicted by Alphafold2, conformation alignment of FL9 and Qdm (Figure 6. a, b) and the NetMHCpan prediction of Qa1b binding of Qdm, FL9 and various FL9 mutant peptides (Supplementary Fig. 8 c) to help readers visualize the reasoning behind our strategy.

References

Azzam, H.S., DeJarnette, J.B., Huang, K., Emmons, R., Park, C.S., Sommers, C.L., El-Khoury, D.,Shores, E.W., and Love, P.E. (2001). Fine tuning of TCR signaling by CD5. J Immunol 166, 5464-5472.10.4049/jimmunol.166.9.5464, PMID:11313384

Fulton, R.B., Hamilton, S.E., Xing, Y., Best, J.A., Goldrath, A.W., Hogquist, K.A., and Jameson, S.C. (2015). The TCR's sensitivity to self peptide-MHC dictates the ability of naive CD8(+) T cells to respond to foreign antigens. Nat Immunol 16, 107-117.10.1038/ni.3043, PMID:25419629

Valerio, M.M., Arana, K., Guan, J., Chan, S.W., Yang, X., Kurd, N., Lee, A., Shastri, N., Coscoy, L., and Robey, E.A. (2023). The promiscuous development of an unconventional Qa1b-restricted T cell population. bioRxiv, 2022.2009.2026.509583.10.1101/2022.09.26.509583,